# The Many Faces of Adversarial Risk

**Muni Sreenivas Pydi**
University of Wisconsin - Madison
Madison, Wisconsin, USA
`pydi@wisc.edu`

**Varun Jog**
University of Cambridge
Cambridge, UK
`vj270@cam.ac.uk`

## Abstract

Adversarial risk quantifies the performance of classifiers on adversarially perturbed data. Numerous definitions of adversarial risk—not all mathematically rigorous and differing subtly in the details—have appeared in the literature. In this paper, we revisit these definitions, make them rigorous, and critically examine their similarities and differences. Our technical tools derive from optimal transport, robust statistics, functional analysis, and game theory. Our contributions include the following: generalizing Strassen's theorem to the unbalanced optimal transport setting with applications to adversarial classification with unequal priors; showing an equivalence between adversarial robustness and robust hypothesis testing with $\infty$-Wasserstein uncertainty sets; proving the existence of a pure Nash equilibrium in the two-player game between the adversary and the algorithm; and characterizing adversarial risk by the minimum Bayes error between a pair of distributions belonging to the $\infty$-Wasserstein uncertainty sets. Our results generalize and deepen recently discovered connections between optimal transport and adversarial robustness and reveal new connections to Choquet capacities and game theory.

## 1   Introduction

Neural networks are known to be vulnerable to *adversarial attacks*, which are imperceptible perturbations to input data that maximize loss [38, 15, 5]. Developing algorithms resistant to such attacks has received considerable attention in recent years [8, 28, 24, 20], motivated by safety-critical applications such as autonomous driving [18, 27], medical imaging [17, 23, 22] and law [21, 6].

A classification algorithm with high accuracy (low risk) in the absence of an adversary may have poor accuracy (high risk) when an adversary is present. Thus, a modified notion known as *adversarial risk* is used to quantify the adversarial robustness of algorithms. Algorithms that minimize adversarial risk are deemed robust. Procedures for finding them have been effective in practice [24, 41, 28], spurring numerous theoretical investigations into adversarial risk and its minimizers.

There is no universally agreed upon definition of adversarial risk. Even the simplest setting of binary classification in $\mathbb{R}^d$ with an $\ell_2$ adversary admits various definitions involving set expansions [9, 16], transport maps [29], Markov kernels [31], and couplings [26]. These works broadly interpret adversarial risk as a measure of robustness to small perturbations, but their definitions differ in subtle details such as the class of adversaries and algorithms considered, budget constraints placed on the adversary, assumptions on the loss function, and the geometry of decision boundaries.

*Optimal adversarial risk* is most commonly defined as the minimax risk under adversarial contamination [24, 33]. Other notable characterizations include an optimal transport cost between data generating distributions in [30, 2, 10, 11], the optimal value of a distributionally robust optimization problem [36, 35, 40], and the value of a two-player zero-sum game [26, 29, 3, 4].

The diversity of definitions for adversarial risk makes it challenging to compare approaches. Moreover, not all approaches are rigorous. For instance, the classes of adversarial strategies and classifier

35th Conference on Neural Information Processing Systems (NeurIPS 2021).

algorithms are often unclear, and issues of measurability are ignored. Although this may be harmless for applied research, it has led to incorrect proofs and insufficient assumptions in some theoretical works; a mathematically rigorous foundation for adversarial risk is essential for future research.

In this paper, we examine various notions of adversarial risk for binary classification in a non-parametric setting, where the decision boundary (or decision region) of a classifier is an arbitrary subset of the input space. We present rigorous definitions of adversarial risk and identify conditions under which these definitions are equivalent. We consider the general setting of Polish spaces (complete, separable metric spaces), and present stronger results for the Euclidean space ($\mathbb{R}^d$). Our contributions are as follows:

- We examine the definition of adversarial risk based on set expansions. For Polish spaces, we observe that adversarial risk is not Borel measurable, and hence, not well-defined when the decision region is an arbitrary set. We show that the problem can be resolved by considering a Polish space equipped with the universal completion of the Borel $\sigma$-algebra and restricting the decision regions to Borel sets. For the Euclidean space with the Lebesgue $\sigma$-algebra, we show that adversarial risk is well-defined for any Lebesgue measurable decision region. Our key lemma (Lemma 4.3) shows that the Lebesgue $\sigma$-algebra is preferred over the Borel $\sigma$-algebra because set expansions are Lebesgue measurable but not necessarily Borel measurable. These results are contained in Section 4.

- We show that the definition of adversarial risk using set expansions is identical to a notion of risk that appears in robust hypothesis testing with $\infty$-Wasserstein uncertainty sets. We prove this result in Polish spaces using the theory of measurable selections [1, 43]. In $\mathbb{R}^d$, we are able to use the powerful theory of Choquet capacities [7] (in particular, Huber and Strassen's 2-alternating capacities [19]) to establish results of a similar nature. These results are contained in Section 5.

- We consider the binary classification setup with unequal priors and show (under suitable assumptions) that the optimal adversarial risk from the above definitions is characterized by an unbalanced optimal transport cost between data-generating distributions. For both Polish spaces and $\mathbb{R}^d$, the main tool we use is Theorem 6.1 in which we generalize a classical result of Strassen on excess-cost optimal transport [37, 42] from probability measures to finite measures with possibly unequal mass. This generalizes results of [31, 2] on binary classification, which were only for equal priors. These results are contained in Section 6.

- We consider the setup of a zero-sum game between the adversary and the algorithm. We show that the value of this game (adversarial risk) is equal to the minimum Bayes error between a pair of distributions belonging to the $\infty$-Wasserstein uncertainty sets centered around true data-generating distributions. We prove the existence of a pure Nash equilibrium in this game for $\mathbb{R}^d$ and for Polish spaces with a *midpoint property*. This extends/strengthens the results of [26, 29, 3] to non-parametric classifiers. These results are contained in Section 7.

The paper is organized as follows: In Section 2, we present preliminary definitions from optimal transport and metric space topology. In Section 3, we discuss various definitions of adversarial risk and present more related work. Sections 4, 5, 6 and 7 contain our main contributions summarized above. We conclude the paper in Section 8 and discuss future research directions.

We emphasize that rectifying measure theoretic issues with existing formulations of adversarial risk is one of our contributions, but not the main focus of our paper. We start our presentation with fixing measurability and well-definedness (in Section 4) because otherwise we will not be able to rigorously present our main results in the subsequent sections, namely: relation to robust hypothesis testing and Choquet capacities in section 5, generalizing the results of [2, 30] in section, 6 proving minimax theorems and existence of Nash equilibria and extending the results of [26, 3, 29] in section 7.

**Notation:** Throughout the paper, we use $\mathcal{X}$ to denote a Polish space (a complete, separable metric space) with metric $d$ and Borel $\sigma$-algebra $\mathcal{B}(\mathcal{X})$. For $x \in \mathcal{X}$ and $r \geq 0$, let $B_r(x)$ denote the closed ball of radius $r$ centered at $x$. We use $\mathcal{P}(\mathcal{X})$ and $\mathcal{M}(\mathcal{X})$ to denote the space of probability measures and finite measures defined on the measure space $(\mathcal{X}, \mathcal{B}(\mathcal{X}))$, respectively. Let $\overline{\mathcal{B}}(\mathcal{X})$ denote the universal completion of $\mathcal{B}(\mathcal{X})$. Let $\overline{\mathcal{P}}(\mathcal{X})$ and $\overline{\mathcal{M}}(\mathcal{X})$ denote the space of probability measures and finite measures defined on the complete measure space $(\mathcal{X}, \overline{\mathcal{B}}(\mathcal{X}))$. For $\mu, \nu \in \mathcal{M}(\mathcal{X})$, we say $\nu$ *dominates* $\mu$ if $\mu(A) \leq \nu(A)$ for all $A \in \mathcal{B}(\mathcal{X})$ and write $\mu \preceq \nu$. When $\mathcal{X}$ is $\mathbb{R}^d$, we use $\mathcal{L}(\mathcal{X})$ to

denote the Lebesgue $\sigma$-algebra and $\lambda$ to denote the $d$-dimensional Lebesgue measure. Note that $\mathcal{L}(\mathcal{X}) = \overline{\mathcal{B}}(\mathcal{X})$ for $\mathcal{X} = \mathbb{R}^d$. For a positive integer $n$, we use $[n]$ to denote the finite set $\{1, \ldots, n\}$.

## 2 Preliminaries

### 2.1 Metric Space Topology

We introduce three different notions of set expansions. For $\epsilon \geq 0$ and $A \in \mathcal{B}(\mathcal{X})$, the $\epsilon$-*Minkowski expansion* of $A$ is given by $A^{\oplus \epsilon} := \cup_{a \in A} B_\epsilon(a)$. The $\epsilon$-*closed expansion* of $A$ is defined as $A^\epsilon := \{x \in \mathcal{X} : d(x, A) \leq \epsilon\}$, where $d(x, A) = \inf_{a \in A} d(x, a)$. The $\epsilon$-*open expansion* of $A$ is defined as $A^{\epsilon)} := \{x \in \mathcal{X} : d(x, A) < \epsilon\}$. We use the notation $A^{-\epsilon}$ to denote $((A^c)^\epsilon)^c$. Similarly, $A^{\ominus \epsilon} := ((A^c)^{\oplus \epsilon})^c$. For example, consider the set $A = (0, 1]$ in the space $(\mathcal{X}, d) = (\mathbb{R}, |\cdot|)$ and $\epsilon > 0$. Then $A^{\oplus \epsilon} = (-\epsilon, 1 + \epsilon]$, $A^\epsilon = [-\epsilon, 1 + \epsilon]$ and $A^{\epsilon)} = (-\epsilon, 1 + \epsilon)$. For any $A \in \mathcal{B}(\mathcal{X})$, $A^\epsilon$ is closed and $A^{\epsilon)}$ is open. Hence, $A^\epsilon, A^{\epsilon)} \in \mathcal{B}(\mathcal{X})$. Moreover, $A^{\epsilon)} \subseteq A^{\oplus \epsilon} \subseteq A^\epsilon$. However, $A^{\oplus \epsilon}$ may not be in $\mathcal{B}(\mathcal{X})$ (see Appendix for an example). In general, the Minkowski sum of two Borel sets need not be Borel [13], and that of two Lebesgue measurable sets need not be Lebesgue measurable [34].

### 2.2 Optimal Transport

Let $\mu, \nu \in \mathcal{P}(\mathcal{X})$. A *coupling* between $\mu$ and $\nu$ is a joint probability measure $\pi \in \mathcal{P}(\mathcal{X}^2)$ with marginals $\mu$ and $\nu$. The set $\Pi(\mu, \nu) \subseteq \mathcal{P}(\mathcal{X}^2)$ denotes the set of all couplings between $\mu$ and $\nu$. The *optimal transport cost* between $\mu$ and $\nu$ under a cost function $c : \mathcal{X} \times \mathcal{X} \to [0, \infty)$ is defined as $\mathcal{T}_c(\mu, \nu) = \inf_{\pi \in \Pi(\mu, \nu)} \int_{\mathcal{X}^2} c(x, x') d\pi(x, x')$. For a positive integer $p$, the $p$-*Wasserstein distance* between $\mu$ and $\nu$ is defined as, $W_p(\mu, \nu) = (\mathcal{T}_{d^p}(\mu, \nu))^{\frac{1}{p}}$. The $\infty$-*Wasserstein metric* is defined as $W_\infty(\mu, \nu) = \lim_{p \to \infty} W_p(\mu, \nu)$. It can also be expressed in the following ways:

$$W_\infty(\mu, \nu) = \inf_{\pi \in \Pi(\mu, \nu)} \operatorname*{ess\,sup}_{(x, x') \sim \pi} d(x, x') = \inf\{\delta > 0 : \mu(A) \leq \nu(A^\delta) \forall A \in \mathcal{B}(\mathcal{X})\}. \tag{1}$$

Given a $\mu \in \mathcal{P}(\mathcal{X})$ and a measurable function $f : \mathcal{X} \to \mathcal{X}$, the *push-forward* of $\mu$ by $f$ is defined as a probability measure $f_{\sharp \mu} \in \mathcal{P}(\mathcal{X})$ given by $f_{\sharp \mu} = \mu(f^{-1}(A))$ for all $A \in \mathcal{B}(\mathcal{X})$.

## 3 Adversarial Risk: Definitions and Related Work

We consider a binary classification setting on feature space $\mathcal{X}$. Let $p_0, p_1 \in \mathcal{P}(\mathcal{X})$ be the data-generating distributions for labels 0 and 1, respectively. Let the prior probabilities for labels 0 and 1 be in the ratio $T : 1$ where we assume $T \geq 1$ without loss of generality. For a space of classifiers parametrized by $w \in \mathcal{W}$ and a loss function $\ell : (\mathcal{X} \times \mathcal{Y}) \times \mathcal{W} \to [0, \infty)$, the adversarial risk of a classifier $w \in \mathcal{W}$ under an adversarial budget of $\epsilon \geq 0$ is defined as [24, 33],

$$R_{\oplus \epsilon}(\ell, w) = \mathbb{E}_{(x, y)} \left[ \sup_{d(x, x') \leq \epsilon} \ell((x', y), w) \right]. \tag{2}$$

If the loss function $\ell(\cdot, w)$ is measurable, upper semi-continuous and bounded above for all $w \in \mathcal{W}$, [26] show that $R_\epsilon(\ell, w)$ is well-defined. But in general, it may not be so. A case of special interest is the 0-1 loss function with non-parametric classifiers of the form $f_A(x) := \mathbb{1}\{x \in A\}$ where $A \in \mathcal{B}(\mathcal{X})$. In this case, $\ell_{0/1}((x, y), A) = \mathbb{1}\{x \in A, y = 0\} + \mathbb{1}\{x \in A^c, y = 1\}$. Hence,

$$R_{\oplus \epsilon}(\ell_{0/1}, A) = \frac{T}{T + 1} \mathbb{E}_{p_0} \left[ \sup_{d(x, x') \leq \epsilon} \mathbb{1}\{x' \in A\} \right] + \frac{1}{T + 1} \mathbb{E}_{p_1} \left[ \sup_{d(x, x') \leq \epsilon} \mathbb{1}\{x' \in A^c\} \right]$$

$$= \frac{T}{T + 1} p_0(A^{\oplus \epsilon}) + \frac{1}{T + 1} p_1((A^c)^{\oplus \epsilon}). \tag{3}$$

A problem with the formulation in equation 3 is the ambiguity over the measurability of the sets $A^{\oplus \epsilon}$ and $(A^c)^{\oplus \epsilon}$. Even when $A \in \mathcal{B}(\mathcal{X})$, it is not guaranteed that $A^{\oplus \epsilon}, (A^c)^{\oplus \epsilon} \in \mathcal{B}(\mathcal{X})$ (see Appendix for an example). Hence, $R_{\oplus \epsilon}(\ell_{0/1}, A)$ is not well-defined for all $A \in \mathcal{B}(\mathcal{X})$. It is shown in [31] that $R_{\oplus \epsilon}(\ell_{0/1}, A)$ is well-defined when $A$ is either closed or open, but its validity beyond that is unknown.

A simple fix to this measurability problem is to use closed set expansion instead of the Minkowski set expansion, as done in [25]. This leads to the following formulation.

$$R_\epsilon(\ell_{0/1}, A) = \frac{T}{T+1} p_0(A^\epsilon) + \frac{1}{T+1} p_1((A^c)^\epsilon). \tag{4}$$

The above definition is well-defined for any $A \in \mathcal{B}(\mathcal{X})$ because $A^\epsilon$ and $(A^c)^\epsilon$ are both closed and hence, measurable. However, under the above definition, a point $x \in A$ may be perturbed to $x' \in A^\epsilon$ such that $d(x, x') > \epsilon$. For example, when $A = (0, 1)$, we have $A^\epsilon = [-\epsilon, \epsilon]$ and an adversary may transport $x = \delta > 0$ to $x' = -\epsilon$, violating the budget constraint at $x$.

**Remark 1.** The formulations in equations (2), (3) and (4) can give a strictly positive adversarial risk even for a "perfect" (i.e. Bayes optimal) classifier. This is consistent with the literature on adversarial examples where even a perfect classifier is forced to make errors in the presence of evasion attacks. These formulations of adversarial risk correspond to "constant-in-the-ball" risk of [16] and "corrupted-instance" risk in [9, 25]. Here, an adversarial risk of zero is only possible if the supports of $p_0$ and $p_1$ are non-overlapping and separated by at least $2\epsilon$. This is not the case with other formulations of adversarial risk such as "exact-in-the-ball" risk [16], "prediction-change risk and "error-region" risk [9, 25]. We focus on the "corrupted-instance" family of risks in this work.

Another approach to defining adversarial risk is by explicitly defining the class of adversaries of budget $\epsilon$ as measurable transport maps $f : \mathcal{X} \to \mathcal{X}$ that push-forward the true data distribution such that no point is transported by more than a distance of $\epsilon$; i.e., $d(x, f(x)) \leq \epsilon$. The transport map-based adversarial risk [29] is formally defined as follows:

$$R_{F_\epsilon}(\ell_{0/1}, A) = \sup_{\substack{f_0, f_1 : \mathcal{X} \to \mathcal{X} \\ \forall x \in \mathcal{X}, d(x, f_i(x)) \leq \epsilon}} \frac{T}{T+1} f_{0\sharp p_0}(A) + \frac{1}{T+1} f_{1\sharp p_1}((A^c)). \tag{5}$$

Yet another definition uses the robust hypothesis testing framework with $W_\infty$ uncertainty sets. In this approach, an adversary perturbs the true distribution $p_i$ to a corrupted distribution $p_i'$ such that $W_\infty(p_i, p_i') \leq \epsilon$. From (1), this is equivalent to the existence of a coupling $\pi \in \Pi(p_i, p_i')$ such that $\mathrm{ess\,sup}_{(x, x') \sim \pi} d(x, x') \leq \epsilon$. The adversarial risk with such an adversary is given by

$$R_{\Gamma_\epsilon}(\ell_{0/1}, A) = \sup_{W_\infty(p_1, p_1'), W_\infty(p_0, p_0') \leq \epsilon} \frac{T}{T+1} p_0'(A) + \frac{1}{T+1} p_1'((A^c)). \tag{6}$$

Clearly, $R_{F_\epsilon}(\ell_{0/1}, A) \leq R_{\Gamma_\epsilon}(\ell_{0/1}, A)$, but conditions for equality were not studied in prior work. Moreover, their relation to set expansion-based definitions in (3) and (4) was also unknown.

Next we discuss some characterizations of optimal adversarial risk, defined as $R^*_{\oplus\epsilon} := \inf_{A \in \mathcal{B}(\mathcal{X})} R_{\oplus\epsilon}(\ell_{0/1}, A)$. In [30, 2], it is shown that $R^*_\epsilon = \frac{1}{2}[1 - D_\epsilon(p_0, p_1)]$ for equal priors ($T = 1$), where $D_\epsilon$ is an optimal transport cost defined as follows.

**Definition 1** ($D_\epsilon$ cost). Let $\mu, \nu \in \mathcal{P}(\mathcal{X})$. Let $\epsilon \geq 0$. Let $c_\epsilon : \mathcal{X}^2 \to \{0, 1\}$ be such that $c_\epsilon(x, x') = \mathbb{1}\{(x, x') \in \mathcal{X} \times \mathcal{X} : d(x, x') > 2\epsilon\}$. Then for $\mu, \nu \in \mathcal{P}(\mathcal{X})$ and $\epsilon \geq 0$, $D_\epsilon(\mu, \nu) = \mathcal{T}_{c_\epsilon}(\mu, \nu)$.

For $\epsilon = 0$, $D_\epsilon$ reduces to the total variation distance. While $D_0$ is a metric on $\mathcal{P}(\mathcal{X})$, $D_\epsilon$ (for $\epsilon > 0$) is neither a metric nor a pseudometric [31].

Other formulations of optimal adversarial risk are inspired from game theory [29, 26, 3]. Consider a game between two players: (1) The adversary whose action space is pairs of distributions $p_0', p_1' \in \overline{\mathcal{P}}(\mathcal{X})$, and (2) The algorithm whose action space is the space of decision regions of the form $A \in \mathcal{B}(\mathcal{X})\}$. For $T > 0$, define $r : \mathcal{B}(\mathcal{X}) \times \overline{\mathcal{P}}(\mathcal{X}) \times \overline{\mathcal{P}}(\mathcal{X}) \to [0, 1]$ as, $r(A, \mu, \nu) = \frac{T}{T+1} \mu(A) + \frac{1}{T+1} \nu((A^c))$. The payoff when the algorithm plays first is given by $\inf_{A \in \mathcal{B}(\mathcal{X})} \sup_{W_\infty(p_0, p_0'), W_\infty(p_1, p_1') \leq \epsilon} r(A, p_0', p_1')$, and this quantity is interpreted as the optimal adversarial risk in this setup.

## 4 Well-Definedness of Adversarial Risk

As stated in Section 3, $R_{\oplus\epsilon}(\ell_{0/1}, A)$ may not be well-defined for some decision regions $A \in \mathcal{B}(\mathcal{X})$ because of the non-measurability of the sets $A^{\oplus\epsilon}$ and $(A^c)^{\oplus\epsilon}$. Specifically, we have the following lemma.

Table 1: Summary of the adversarial risk definitions presented in Section 3. $R_{\oplus\epsilon}, R_\epsilon, R_{F_\epsilon}$ and $R_{\Gamma_\epsilon}$ denote adversarial risk defined using Minkowski set expansions, closed set expansions, transport maps and $\infty$-Wasserstein metric respectively.

| Risk | Adversary's action | Perturbation | $d(x, x') \leq \epsilon$? | Well-defined? |
|---|---|---|---|---|
| $R_{\oplus\epsilon}$ | $x \in A \rightarrow x' \in A^{\oplus\epsilon}$ | Random | Yes, $\forall x$ | $\mathbb{R}^d$: Yes, for Lebesgue $A$ 
 Polish $\mathcal{X}$: Yes, for Borel $A$ |
| $R_\epsilon$ | $x \in A \rightarrow x' \in A^\epsilon$ | Random | No | Yes, for measurable $A$ |
| $R_{F_\epsilon}$ | $x \rightarrow x' = f_i(x)$ | Deterministic | Yes, $\forall x$ | Yes, for measurable $f_i$ |
| $R_{\Gamma_\epsilon}$ | $p_i \rightarrow p'_1$ 
 $W_\infty(p_i, p'_i) \leq \epsilon$ | Random | Almost surely yes, $\forall x$ | Yes |

**Lemma 4.1.** For any $\epsilon > 0$, there exists $A \in \mathcal{B}(\mathcal{X})$ such that $A^{\oplus\epsilon} \notin \mathcal{B}(\mathcal{X})$.

In this section, we lay down the conditions under which the ambiguity can be resolved. We begin by presenting a Lemma that shows that $A^{\oplus\epsilon}$ is an analytic set (i.e. a continuous image of a Borel set) whenever $A$ is Borel. It is known that an analytic sets are universally measurable, i.e. they belong in $\overline{\mathcal{B}}(\mathcal{X})$, the universal completion of the Borel $\sigma$-algebra $\mathcal{B}(\mathcal{X})$, and are measurable with respect to any finite measure defined on the complete measure space, $(\mathcal{X}, \overline{\mathcal{B}}(\mathcal{X}))$.

**Lemma 4.2.** Let $A \in \mathcal{B}(\mathcal{X})$. Then, $A^{\oplus\epsilon}$ is an analytic set. Consequently, $A^{\oplus\epsilon} \in \overline{\mathcal{B}}(\mathcal{X})$.

By virtue of the previous lemma, we have the following.

**Theorem 4.1.** Let $p_0, p_1 \in \overline{\mathcal{P}}(\mathcal{X})$. Then for any $A \in \mathcal{B}(\mathcal{X})$, $R_{\oplus\epsilon}(\ell_{0/1}, A)$ is well-defined.

For the special case of $\mathcal{X} = \mathbb{R}^d$, we can further strengthen Theorem 4.1 to include all Lebesgue measurable sets $\mathcal{L}(\mathcal{X})$ instead of just Borel sets $\mathcal{B}(\mathcal{X})$. For this, we use the concept of porous sets.

**Definition 2** (Porous set). A set $E \subseteq \mathcal{X}$ is said to be porous if there exists $\alpha \in (0, 1)$ and $r_0 > 0$ such that for every $r \in (0, r_0]$ and every $x \in \mathcal{X}$, there is an $x' \in \mathcal{X}$ such that $B_{\alpha r}(x') \subseteq B_r(x) \backslash E$.

Porous sets are a subclass of nowhere dense sets. Importantly, $\lambda(E) = 0$ for any porous set $E \subseteq \mathbb{R}^d$ [47]. By the following lemma, the set difference between the closed/open set expansions is porous.

**Lemma 4.3.** Let $(\mathcal{X}, d) = (\mathbb{R}^d, \|\cdot\|)$ and $A \in \mathcal{L}(\mathcal{X})$. Then $E = A^\epsilon \backslash A^\epsilon)$ is porous.

Lemma 4.3 plays a crucial role in proving that $A^{\oplus\epsilon} \in \mathcal{L}(\mathcal{X})$ whenever $A \in \mathcal{L}(\mathcal{X})$. We recall that $A^{\oplus\epsilon}$ is the Minkowski sum of $A$ with the closed $\epsilon$-ball. In general, the Minkowski sum of two Lebesgue measurable sets is not always Lebesgue measurable [34, 14]. So the fact that one of them is a closed ball in case of $A^{\oplus\epsilon}$ is important. In the following theorem, we use Lemma 4.3 to prove the measurability of $A^{\oplus\epsilon}$ and in turn prove that $R_{\oplus\epsilon}(\ell_{0/1}, A)$ is well-defined for any $A \in \mathcal{L}(\mathcal{X})$.

**Theorem 4.2.** Let $(\mathcal{X}, d) = (\mathbb{R}^d, \|\cdot\|)$. Let $p_0, p_1 \in \overline{\mathcal{P}}(\mathcal{X})$ and let $\epsilon \geq 0$. Then for any $A \in \mathcal{L}(\mathcal{X})$, $R_{\oplus\epsilon}(\ell_{0/1}, A)$ is well-defined. If, in addition, $p_0$ and $p_1$ are absolutely continuous with respect to the Lebesgue measure, then $R_{\oplus\epsilon}(\ell_{0/1}, A) = R_\epsilon(\ell_{0/1}, A)$.

# 5 Equivalence with $\infty$-Wasserstein Robustness

In this section, we show the conditions under which $R_{\oplus\epsilon}(\ell_{0/1}, A)$ is equivalent to other notions of adversarial risk based on transport maps and $W_\infty$ robustness.

## 5.1 $W_\infty$ Robustness in Polish Spaces via Measurable Selections

We begin by presenting a lemma that links the measure of $\epsilon$-Minkowsi set expansion to the worst case measure over a $W_\infty$ probability ball of radius $\epsilon$.

**Lemma 5.1.** Let $\mu \in \overline{\mathcal{P}}(\mathcal{X})$ and $A \in \mathcal{B}(\mathcal{X})$. Then $\sup_{W_\infty(\mu, \mu') \leq \epsilon} \mu'(A) = \mu(A^{\oplus\epsilon})$. Moreover, the supremum in the previous equation is achieved by a $\mu^* \in \mathcal{P}(\mathcal{X})$ that is induced from $\mu$ via a measurable transport map $\phi : \mathcal{X} \rightarrow \mathcal{X}$ (i.e. $\mu^* = \phi_{\sharp}\mu$) satisfying $d(x, \phi(x)) \leq \epsilon$ for all $x \in \mathcal{X}$.

Table 2: Eqvivalences among adversarial risk formulations. $R_{\oplus\epsilon}(A), R_\epsilon(A), R_{F_\epsilon}(A)$ and $R_{\Gamma_\epsilon}(A)$ denote adversarial risk for 0-1 loss function ($\ell_{0/1}$) for a binary classifier with decision region $A$ (i.e. $f_A(x) = \mathbb{1}\{x \in A\}$), defined using Minkowski set expansions, closed set expansions, transport maps and $\infty$-Wasserstein metric respectively. $\mathcal{B}(\mathcal{X})$ and $\mathcal{L}(\mathcal{X})$ denote the Borel and Lebesgue $\sigma$-algebras. $(\mathcal{X}, \overline{\mathcal{B}}(\mathcal{X}))$ denotes the universal completion of the Borel measure space, $(\mathcal{X}, \mathcal{B}(\mathcal{X}))$.

| Equivalences in Adversarial Risk | Conditions |
|---|---|
| $R_{\oplus\epsilon}(A) = R_{\Gamma_\epsilon}(A)$ | $\mathbb{R}^d$: $A \in \mathcal{L}(\mathcal{X})$ or $(\mathcal{X}, \overline{\mathcal{B}}(\mathcal{X}))$: $A \in \mathcal{B}(\mathcal{X})$ |
| $R_{\oplus\epsilon}(A) = R_{\Gamma_\epsilon}(A) = R_{F_\epsilon}(A)$ | $(\mathcal{X}, \overline{\mathcal{B}}(\mathcal{X}))$: $A \in \mathcal{B}(\mathcal{X})$ |
| $R_{\oplus\epsilon}(A) = R_{\Gamma_\epsilon}(A) = R_{F_\epsilon}(A) = R_\epsilon(A)$ | $\mathbb{R}^d$: $A \in \mathcal{L}(\mathcal{X})$ and $p_0, p_1$ have densities |

A crucial step in the proof of Lemma 5.1 is finding a measurable transport map $\phi$ such that $\phi^{-1}(A) = A^{\oplus\epsilon}$ and $d(x, \phi(x)) \leq \epsilon$ for all $x \in \mathcal{X}$. In the following theorem, we use Lemma 5.1 to establish the equivalence between three different notions of adversarial risk introduced in section 3.

**Theorem 5.1.** Let $p_0, p_1 \in \overline{\mathcal{P}}(\mathcal{X})$ and $A \in \mathcal{B}(\mathcal{X})$. Then $R_{\oplus\epsilon}(\ell_{0/1}, A) = R_{F_\epsilon}(\ell_{0/1}, A) = R_{\Gamma_\epsilon}(\ell_{0/1}, A)$. In addition, the supremum over $f_0$ and $f_1$ in $R_{F_\epsilon}(\ell_{0/1}, A)$ is attained. Similarly, the supremum over $p_0'$ and $p_1'$ in $R_{\Gamma_\epsilon}(\ell_{0/1}, A)$ is attained.

## 5.2 $W_\infty$ Robustness in $\mathbb{R}^d$ via 2-Alternating Capacities

In this subsection, we establish a connection between adversarial risk and Choquet capacities [7] in $\mathbb{R}^d$. This connection allows us to extend Theorem 5.1 from Borel sets to the broader class of Lebesgue measurable sets. We will again use this connection for proving minimax theorems and existence of Nash equilibria in Section 7.1. We begin with the following definitions.

**Definition 3** (Capacity). A set function $v : \mathcal{B}(\mathcal{X}) \to [0, 1]$ is a *capacity* if it satisfies the following conditions: (1) $v(\varnothing) = 0$ and $v(\mathcal{X}) = 1$; (2) For $A, B \in \mathcal{B}(\mathcal{X})$, $A \subseteq B \implies v(A) \leq v(B)$; (3) $A_n \uparrow A \implies v(A_n) \uparrow v(A)$; and (4) $F_n \downarrow F$, $F_n$ closed $\implies v(F_n) \downarrow v(F)$.

**Definition 4** (2-Alternating Capacity). A capacity $v$ defined on the measure space $(\mathcal{X}, \mathcal{B}(\mathcal{X}))$ is called 2-alternating if $v(A \cup B) + v(A \cap B) \leq v(A) + v(B)$ for all $A, B \in \mathcal{B}(\mathcal{X})$.

For any compact set of probability measures $\Xi \subseteq \mathcal{P}(\mathcal{X})$, the upper probability $v(A) = \sup_{\mu \in \Xi} \mu(A)$ is a capacity [19]. The upper probability of $\epsilon$-neighborhoods of a $\mu \in \mathcal{P}(\mathcal{X})$ defined using either the total variation metric or the Levy-Prokhorov metric can be shown to be a 2-alternating capacity [19]. The following lemma shows that $A \mapsto \mu(A^{\oplus\epsilon})$ is a 2-alternating capacity under some conditions.

**Lemma 5.2.** Let $(\mathcal{X}, d) = (\mathbb{R}^d, \|\cdot\|)$. Let $\mu \in \overline{\mathcal{P}}(\mathcal{X})$ and let $\epsilon \geq 0$. Define a set function $v$ on $\mathcal{X}$ such that for any $A \in \mathcal{L}(\mathcal{X})$, $v(A) := \mu(A^{\oplus\epsilon})$. Then $v$ is a 2-alternating capacity.

Now we relate the capacity defined in Lemma 5.2 to the $W_\infty$ metric. Since the $\epsilon$-neighborhood of a $\mu \in \mathcal{P}(\mathcal{X})$ in $W_\infty$ metric is a compact set of probability measures [46], the upper probability over this $W_\infty$ $\epsilon$-ball is a capacity. The following lemma shows that it is a 2-alternating capacity, and identifies it with the capacity defined in Lemma 5.2.

**Lemma 5.3.** Let $(\mathcal{X}, d) = (\mathbb{R}^d, \|\cdot\|)$. Let $\mu \in \overline{\mathcal{P}}(\mathcal{X})$. Then for any $A \in \mathcal{L}(\mathcal{X})$, $\sup_{W_\infty(\mu, \mu') \leq \epsilon} \mu'(A) = \mu(A^{\oplus\epsilon})$. Moreover, the supremum in the previous equation is attained.

Lemma 5.3 plays a similar role to Lemma 5.1 in proving the following equivalence between adversarial robustness and $W_\infty$ robustness.

**Theorem 5.2.** Let $(\mathcal{X}, d) = (\mathbb{R}^d, \|\cdot\|)$. Let $p_0, p_1 \in \overline{\mathcal{P}}(\mathcal{X})$ and let $\epsilon \geq 0$. Then for any $A \in \mathcal{L}(\mathcal{X})$, $R_{\oplus\epsilon}(\ell_{0/1}, A) = R_{\Gamma_\epsilon}(\ell_{0/1}, A)$, and the supremum over $p_0'$ and $p_1'$ in $R_{\Gamma_\epsilon}(\ell_{0/1}, A)$ is attained.

The proof follows by converting the expression for $R_{\Gamma_\epsilon}$ into one for $R_{\oplus\epsilon}$ using Lemma 5.3. Unlike Theorem 5.1, Theorem 5.2 does not show the equivalence of $R_{F_\epsilon}(\ell_{0/1}, A)$ with the other definitions under the relaxed assumption of $A \in \mathcal{L}(\mathcal{X})$. This is because Lemma 5.3 does not provide a push-forward map $\phi$ such that $\mu^* = \phi_{\sharp\mu}$ with $\mu^*$ attaining the supremum over the $W_\infty$ ball.

# 6 Optimal Adversarial Risk via Generalized Strassen's Theorem

In section 5, we analyzed adversarial risk for a specific decision region $A \in \mathcal{B}(\mathcal{X})$. In this section, we analyze infimum of adversarial risk over all possible decision regions; i.e., the optimal adversarial risk. We show that optimal adversarial risk in binary classification with unequal priors is characterized by an unbalanced optimal transport cost between data-generating distributions. Our main technical lemma generalizes Strassen's theorem to unbalanced optimal transport. We present this result in subsection 6.1 and present our characterization of optimal adversarial risk in subsection 6.2.

## 6.1 Unbalanced Optimal Transport & Generalized Strassen's Theorem

Recall from Section 3 that the optimal transport cost $D_\epsilon$ characterizes the optimal adversarial risk in binary classification for equal priors. The following result gives an alternative characterization of $D_\epsilon$.

**Proposition 6.1** (Strassen's theorem). [Corollary 1.28 in [42]] Let $\mu, \nu \in \mathcal{P}(\mathcal{X})$. Let $\epsilon \geq 0$. Then

$$\sup_{A \in \mathcal{B}(\mathcal{X})} \mu(A) - \nu(A^{2\epsilon}) = D_\epsilon(\mu, \nu). \tag{7}$$

Proposition 6.1 is a special case of Kantorovich-Rubinstein duality [42] applied to $\{0, 1\}$-valued cost functions. We now generalize this result to measures with unequal masses. We begin with some definitions that generalize the concepts we introduced in subsection 2.2.

Let $\mu, \nu \in \mathcal{M}(\mathcal{X})$ be such that $\mu(\mathcal{X}) \leq \nu(\mathcal{X})$. A *coupling* between $\mu$ and $\nu$ is a measure $\pi \in \mathcal{M}(\mathcal{X}^2)$ such that for any $A \in \mathcal{B}(\mathcal{X})$, $\pi(A \times \mathcal{X}) = \mu(A)$ and $\pi(\mathcal{X} \times A) \leq \nu(A)$. The set $\Pi(\mu, \nu)$ is defined to be the set of all couplings between $\mu$ and $\nu$. For a cost function $c : \mathcal{X}^2 \to [0, \infty)$, the optimal transport cost between $\mu$ and $\nu$ under $c$ is defined as $\mathcal{T}_c(\mu, \nu) = \inf_{\pi \in \Pi(\mu, \nu)} \int_{\mathcal{X}^2} c(x, x') d\pi(x, x')$.

**Theorem 6.1** (Generalized Strassen's theorem). Let $\mu, \nu \in \mathcal{M}(\mathcal{X})$ be such that $0 < M = \mu(\mathcal{X}) \leq \nu(\mathcal{X})$. Let $\epsilon > 0$. Define $c_\epsilon : \mathcal{X}^2 \to \{0, 1\}$ as $c_\epsilon(x, x') = \mathbb{1}\{(x, x') \in \mathcal{X}^2 : d(x, x') > 2\epsilon\}$. Then

$$\sup_{A \in \mathcal{B}(\mathcal{X})} \mu(A) - \nu(A^{2\epsilon}) = \mathcal{T}_{c_\epsilon}(\mu, \nu) = M \inf_{\nu' \in \mathcal{P}(\mathcal{X}):\nu' \preceq \nu/M} D_\epsilon(\mu/M, \nu'). \tag{8}$$

Moreover, the infimum on the right hand side is attained. (Equivalently, there is a coupling $\pi \in \Pi(\mu, \nu)$ that attains the unbalanced optimal transport cost $\mathcal{T}_{c_\epsilon}(\mu, \nu)$.)

Our proof of Theorem 6.1 leverages strong duality in linear programming. We first establish (8) for discrete measures on a finite support. We then apply the discrete result on a sequence of measures supported on a countable dense subset of the Polish space $\mathcal{X}$. Using the tightness of finite measures on $\mathcal{X}$, we construct an optimal coupling that achieves the cost $\mathcal{T}_{c_\epsilon}(\mu, \nu)$ in (8). We then show that the constructed coupling satisfies (8). This proof strategy is adapted from the works of [12] and [32].

## 6.2 Optimal Adversarial Risk for Unequal Priors

Generalized Strassen's theorem involves closed set expansions. The following lemma allows us to switch to Minkowski set expansions.

**Lemma 6.1.** Let $\mu, \nu \in \overline{\mathcal{M}}(\mathcal{X})$ and let $\epsilon \geq 0$. Then $\sup_{A \in \mathcal{B}(\mathcal{X})} \mu(A) - \nu(A^{2\epsilon}) = \sup_{A \in \mathcal{B}(\mathcal{X})} \mu(A^{\ominus \epsilon}) - \nu(A^{\oplus \epsilon})$. Moreover, the supremum in the right hand side of the above equality can be replaced by a supremum over closed sets.

Using the above lemma and the generalized Strassen's theorem, we show the following result on optimal adversarial risk for unequal priors, generalizing the result of [30, 2].

**Theorem 6.2.** Let $p_0, p_1 \in \overline{\mathcal{P}}(\mathcal{X})$ and let $\epsilon \geq 0$. Then,

$$\inf_{A \in \mathcal{B}(\mathcal{X})} R_{\oplus \epsilon}(\ell_{0/1}, A) = \frac{1}{T+1} \left[ 1 - \inf_{q \in \mathcal{P}(\mathcal{X}):q \preceq Tp_0} D_\epsilon(q, p_1) \right]. \tag{9}$$

Moreover, the infimum on the left hand side can be replaced by an infimum over closed sets.

The proof follows by using Lemma 6.1 to convert the expression with Minkowski expansion to one with closed expansions, followed by an application of Theorem 6.1 to arrive at the final optimal transport-based expression. Theorem 6.2 extends the result of [31] in two ways: (1) the infimum is

taken over all sets for which $R_{\oplus\epsilon}(\ell_{0/1}, A)$ is well-defined, instead of restricting to closed sets, and (2) the priors on both labels can be unequal. We also note that for $(\mathcal{X}, d) = (\mathbb{R}^d, \|\cdot\|)$, (9) holds with the infimum on the left hand side taken over all $A \in \mathcal{L}(\mathcal{X})$.

# 7 Minimax Theorems and Nash Equilibria

In this section, we revisit the zero-sum game between the adversary and the algorithm introduced in section 3. Recall that for $A \in \mathcal{B}(\mathcal{X})$ and $p_0', p_1' \in \overline{\mathcal{P}}(\mathcal{X})$, the payoff function is given by

$$r(A, p_0', p_1') = \frac{T}{T+1} p_0'(A) + \frac{1}{T+1} p_1'((A^c)). \tag{10}$$

The max-min inequality gives us

$$\sup_{W_\infty(p_0, p_0'), W_\infty(p_1, p_1') \le \epsilon} \inf_{A \in \mathcal{A}} r(A, p_0', p_1') \le \inf_{A \in \mathcal{B}(\mathcal{X})} \sup_{W_\infty(p_0, p_0'), W_\infty(p_1, p_1') \le \epsilon} r(A, p_0', p_1'). \tag{11}$$

If the inequality in (11) is an equality, we say that the game has zero duality gap, and admits a value equal to either expression in (11). Then there is no advantage to a player making the first move. Our minimax theorems establish such an equality. If in addition to having an equality in (11), there exist $p_0^*, p_1^* \in \mathcal{P}(\mathcal{X})$ that achieve the supremum on the left-hand side and $A^* \in \mathcal{B}(\mathcal{X})$ that achieves the infimum on the right-hand side, we say that $((p_0^*, p_1^*), A^*)$ is a pure Nash equilibrium of the game.

In Section 7.1, we prove the minimax theorem and the existence of a pure Nash equilibrium in $\mathbb{R}^d$ using the theory of 2-alternating capacities [19] and the relation to adversarial risk from Section 5.2. Section 7.2 extends these results to more general Polish spaces with a "midpoint property."

## 7.1 Minimax Theorem in $\mathbb{R}^d$ via 2-Alternating Capacities

The following theorem proves the minimax equality and the existence of a Nash equilibrium for the adversarial robustness game in $\mathbb{R}^d$.

**Theorem 7.1** (Minimax theorem in $\mathbb{R}^d$). Let $(\mathcal{X}, d) = (\mathbb{R}^d, \|\cdot\|)$. Let $p_0, p_1 \in \overline{\mathcal{P}}(\mathcal{X})$ and let $\epsilon \ge 0$. Define $r$ as in (10). Then,

$$\sup_{W_\infty(p_0, p_0'), W_\infty(p_1, p_1') \le \epsilon} \inf_{A \in \mathcal{L}(\mathcal{X})} r(A, p_0', p_1') = \inf_{A \in \mathcal{L}(\mathcal{X})} \sup_{W_\infty(p_0, p_0'), W_\infty(p_1, p_1') \le \epsilon} r(A, p_0', p_1'). \tag{12}$$

Moreover, there exist $p_0^*, p_1^* \in \overline{\mathcal{P}}(\mathcal{X})$ and $A^* \in \mathcal{L}(\mathcal{X})$ that achieve the supremum and infimum on the left and right hand sides of the above equation.

Crucial to the proof of Theorem 7.1 is Lemma 5.2, which shows that the set-valued maps $A \mapsto p_0(A^{\oplus\epsilon})$ and $A^c \mapsto p_1((A^c)^{\oplus\epsilon})$ are 2-alternating capacities. The same proof technique is not applicable in general Polish spaces because the map $A \mapsto \mu(A^{\oplus\epsilon})$ is not a capacity for a general $\mu \in \overline{\mathcal{P}}(\mathcal{X})$. This is because $A^{\oplus\epsilon}$ is not measurable for all $A \in \overline{\mathcal{B}}(\mathcal{X})$.

## 7.2 Minimax Theorem in Polish Spaces via Optimal Transport

We now extend the minimax theorem from $\mathbb{R}^d$ to general Polish spaces with the following property.

**Definition 5** (Midpoint property). A metric space $(\mathcal{X}, d)$ is said to have the midpoint property if for every $x_1, x_2 \in \mathcal{X}$, there exists $x \in \mathcal{X}$ such that, $d(x_1, x) = d(x, x_2) = d(x_1, x_2)/2$.

Any normed vector space with distance defined as $d(x, x') = \|x - x'\|$ satisfies the midpoint property. An example of a metric space without this property is the discrete metric space where $d(x, x') = \mathbb{1}\{x \ne x'\}$. The midpoint property plays a crucial role in proving the following theorem, which shows that the $D_\epsilon$ transport cost between two distributions is the shortest total variation distance between their $\epsilon$-neighborhoods in $W_\infty$ metric. A similar result was also presented in [11].

**Theorem 7.2** ($D_\epsilon$ as shortest $D_{TV}$ between $W_\infty$ balls). Let $(\mathcal{X}, d)$ have the midpoint property. Let $\mu, \nu \in \overline{\mathcal{P}}(\mathcal{X})$ and let $\epsilon \ge 0$. Then $D_\epsilon(\mu, \nu) = \inf_{W_\infty(\mu, \mu'), W_\infty(\nu, \nu') \le \epsilon} D_{TV}(\mu', \nu')$. Moreover, the infimum over $D_{TV}$ in the above equation is attained.

The following theorem uses Theorem 7.2 to prove the minimax equality and the existence of a Nash equilibrium for any Polish space with the midpoint property for the case of equal priors.

**Theorem 7.3** (Minimax theorem for equal priors). Let $(\mathcal{X}, d)$ have the midpoint property. Let $p_0, p_1 \in \overline{\mathcal{P}}(\mathcal{X})$ and let $\epsilon \geq 0$. Define $r$ as in (10) with $T = 1$. Then

$$\sup_{W_\infty(p_0,p_0'), W_\infty(p_1,p_1') \leq \epsilon} \inf_{A \in \mathcal{B}(\mathcal{X})} r(A, p_0', p_1') = \inf_{A \in \mathcal{B}(\mathcal{X})} \sup_{W_\infty(p_0,p_0'), W_\infty(p_1,p_1') \leq \epsilon} r(A, p_0', p_1'). \quad (13)$$

Moreover, there exist $p_0^*, p_1^* \in \mathcal{P}(\mathcal{X})$ that achieve the supremum on the left hand side.

*Proof.* For $\mu \in \overline{\mathcal{P}}(\mathcal{X})$, let $WB(\mu)$ denote the set of all $\mu' \in \overline{\mathcal{P}}(\mathcal{X})$ such that $W_\infty(\mu, \mu') \leq \epsilon$.

$$\inf_{A \in \mathcal{B}(\mathcal{X})} \sup_{\substack{p_0' \in WB(p_0) \\ p_1' \in WB(p_1)}} r(A, p_0', p_1') = \inf_{A \in \mathcal{B}(\mathcal{X})} R_{\Gamma_\epsilon}(\ell_{0/1}, A) \stackrel{(i)}{=} \inf_{A \in \mathcal{B}(\mathcal{X})} R_{\oplus \epsilon}(\ell_{0/1}, A) \stackrel{(ii)}{=} \frac{1}{2} \left[1 - D_\epsilon(p_0, p_1)\right]$$

$$\sup_{\substack{p_0' \in WB(p_0) \\ p_1' \in WB(p_1)}} \inf_{A \in \mathcal{B}(\mathcal{X})} r(A, p_0', p_1') \stackrel{(iii)}{=} \sup_{\substack{p_0' \in WB(p_0) \\ p_1' \in WB(p_1)}} \frac{1}{2} \left[1 - D_{TV}(p_0', p_1')\right] = \frac{1}{2} \left[1 - \inf_{\substack{p_0' \in WB(p_0) \\ p_1' \in WB(p_1)}} D_{TV}(p_0', p_1')\right],$$

where (i) follows from Theorem 5.1, (ii) from Theorem 6.2, and $(iii)$ again from Theorem 6.2 with $\epsilon = 0$. The expressions on the right extremes of the above equations are equal by Theorem 7.2. The existence of $p_0^*, p_1^* \in \overline{\mathcal{P}}(\mathcal{X})$ follows Theorem 7.2. $\qquad \square$

To prove the minimax theorem for unequal priors, we need the following generalization of Theorem 7.2 to finite measures of unequal mass.

**Lemma 7.1.** Let $p_0, p_1 \in \overline{\mathcal{P}}(\mathcal{X})$ and let $\epsilon \geq 0$. Then for $T \geq 1$,

$$\inf_{q \in \overline{\mathcal{P}}(\mathcal{X}): q \preceq T p_0} D_\epsilon(q, p_1) = \inf_{q \in \overline{\mathcal{P}}(\mathcal{X}): q \preceq T p_0} \inf_{W_\infty(q,q'), W_\infty(p_1,p_1') \leq \epsilon} D_{TV}(q', p_1')$$

$$= \inf_{W_\infty(p_0,p_0'), W_\infty(p_1,p_1') \leq \epsilon} \inf_{q' \in \overline{\mathcal{P}}(\mathcal{X}): q' \preceq T p_0'} D_{TV}(q', p_1') \quad (14)$$

Now, we prove the minimax equality for unequal priors.

**Theorem 7.4** (Minimax theorem for unequal priors). Let $(\mathcal{X}, d)$ have the midpoint property. Let $p_0, p_1 \in \overline{\mathcal{P}}(\mathcal{X})$ and let $\epsilon \geq 0$. For $T > 0$, define $r$ as in (10). Then

$$\sup_{W_\infty(p_0,p_0'), W_\infty(p_1,p_1') \leq \epsilon} \inf_{A \in \mathcal{B}(\mathcal{X})} r(A, p_0', p_1') = \inf_{A \in \mathcal{B}(\mathcal{X})} \sup_{W_\infty(p_0,p_0'), W_\infty(p_1,p_1') \leq \epsilon} r(A, p_0', p_1'). \quad (15)$$

The proof uses: (1) the characterization of inf-sup payoff in terms of unbalanced optimal transport using Theorem 5.1; (2) Lemma 7.1; and (3) the minimax equality of Theorem 7.3 for equal priors.

**Remark 2.** Unlike Theorem 7.1, Theorems 7.3 and 7.4 do not guarantee the existence of an optimal decision region $A^*$. While Theorem 7.3 guarantees the existence of worst-case pair of perturbed distributions $p_0^*, p_1^*$, Theorem 7.4 does not do so. Nevertheless, an approximate pure Nash equilibrium exists in all the cases. This is in sharp contrast with the non-existence of Nash equilibrium proven in [29] (which considers a different notion of adversarial perturbations).

**Remark 3.** A recent work [26] shows the existence of mixed Nash equilibrium for randomized classifiers parametrized by points in a Polish space (see also [29, 3]). Fan's minimax theorem used in this result is inapplicable in our setting of non-parametric, decision region-based classifiers. Instead, we applied the theory of Choquet capacities (in $\mathbb{R}^d$) and generalized Strassen's duality theorem (in Polish spaces), which is novel to the best of our knowledge.

## 8 Discussion

We examined different notions of adversarial risk in a binary classification setting with 0-1 loss function and laid down the conditions under which these definitions are equivalent. By verifying the conditions in Sections 4 and 5, researchers may use different definitions interchangeably. Several definitions have also been proposed for adversarial risk under general loss functions [31, 26] using analogous constructions like transport maps, couplings and suprema over $\epsilon$-neighborhoods. Extending our equivalence results to more general loss functions is left for future work.

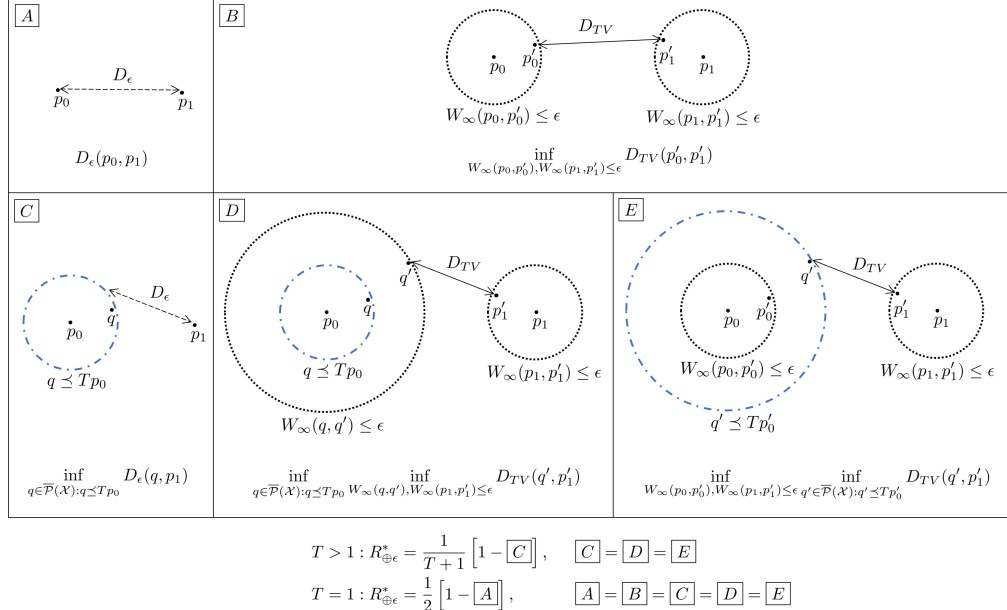

$$T > 1 : R^*_{\oplus\epsilon} = \frac{1}{T+1}\left[1 - \boxed{C}\right], \qquad \boxed{C} = \boxed{D} = \boxed{E}$$

$$T = 1 : R^*_{\oplus\epsilon} = \frac{1}{2}\left[1 - \boxed{A}\right], \qquad \boxed{A} = \boxed{B} = \boxed{C} = \boxed{D} = \boxed{E}$$

Figure 1: Illustration of various equivalent formulations of the optimal adversarial risk. The equalities summarize the results of Section 6 and Section 7. For equal priors ($T = 1$), $\boxed{A}$ and $\boxed{B}$ denote two ways of obtaining the optimal adversarial risk, $R^*_{\oplus\epsilon}$: 1) $\boxed{A}$, which denotes the $D_\epsilon$ cost between the true label distributions $p_0$ and $p_1$, and 2) $\boxed{B}$, which denotes the shortest total variation distance between $\infty$-Wasserstein balls of radius $\epsilon$ around $p_0$ and $p_1$. For unequal priors ($T > 1$), $\boxed{C}$, $\boxed{D}$ and $\boxed{E}$ denote three equivalent ways of obtaining $R^*_{\oplus\epsilon}$. The black dotted balls denote $\infty$-Wasserstein balls and the blue dashed balls denote sets defined using stochastic domination. The order in which the two types of balls appear around $p_0$ is reversed between $\boxed{D}$ and $\boxed{E}$.

We analyzed optimal adversarial risk for (non-parametric) decision region-based classifiers. Using a formulation of optimal transport between finite measures of unequal mass, we extended the optimal transport based characterization of adversarial risk of [30, 2] to unequal priors by generalizing Strassen's theorem. This may find applications in the study of excess cost optimal transport [45, 44]. A recent work [39] obtains a different characterization of optimal adversarial risk using optimal transport on the product space $\mathcal{X} \times \mathcal{Y}$ where $\mathcal{Y}$ is the label space. Further, they show the evolution of the optimal classifier $A^*$ as $\epsilon$ grows, in terms of a mean curvature flow. This raises an interesting question on the evolution of the optimal adversarial distributions $p_0^*, p_1^* \in \overline{\mathcal{P}}(\mathcal{X})$ with $\epsilon$.

We proved a minimax theorem for adversarial robustness game and the existence of a Nash equilibrium. We constructed the worst-case pair of distributions $p_0^*, p_1^* \in \overline{\mathcal{P}}(\mathcal{X})$ in terms of true data distributions and showed that their total variation distance gives the optimal adversarial risk. Identifying worst case distributions could lead to a new approach to developing robust algorithms.

We used Choquet capacities for results in $\mathbb{R}^d$ and measurable selections in Polish spaces. Specifically, we showed that the measure of $\epsilon$-Minkowski expansion is a 2-alternating capacity. This connection could help generalize our results to total variation and Prokhorov distance based contaminations.

**Limitations:** We largely focused on the binary classification setup with 0-1 loss function. While it may be possible extend our results on measurability and relation to $\infty$-Wasserstein distributional robustness to more general loss functions and a multi-class setup, it is unclear how our results on generalized Strassen's theorem and Nash equilibria can be extended further. Our results on various equivalent formulations of optimal adversarial risk are specific to adversarial perturbations (or equivalently, $\infty$-Wasserstein distributional perturbations), and we did not investigate more general perturbation models.

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
