## A  Preliminary Lemmas

**Lemma A.1.** Let $A_n \in \mathcal{B}(\mathcal{X})$ for $n \in \{1, 2, \ldots\}$. Then,

$$(\cup_n A_n)^{\oplus \epsilon} = \cup_n A_n^{\oplus \epsilon},$$
$$(\cap_n A_n)^{\oplus \epsilon} \subseteq \cap_n A_n^{\oplus \epsilon}.$$

*Proof.* Suppose $a \in (\cup_n A_n)^{\oplus \epsilon}$. Then there exists $a_i \in A_i$ for some $i \in \{1, 2, \ldots\}$ such that $d(a, a_i) \leq \epsilon$. Hence, $a \in A_i^{\oplus \epsilon} \subseteq \cup_n A_n^{\oplus \epsilon}$. Therefore, $(\cup_n A_n)^{\oplus \epsilon} \subseteq \cup_n A_n^{\oplus \epsilon}$.

Suppose $b \in \cup_n A_n^{\oplus \epsilon}$. Then $b \in A_j^{\oplus \epsilon}$ for some $j \in \mathbb{N}$. So there must exist $b' \in A_j$ such that $d(b, b') \leq \epsilon$. Since $b' \in \cup_n A_n$, we get that $b \in (\cup_n A_n)^{\oplus \epsilon}$. Therefore, $\cup_n A_n^{\oplus \epsilon} \subseteq (\cup_n A_n)^{\oplus \epsilon}$.

Suppose $c \in (\cap_n A_n)^{\oplus \epsilon}$. Then there exists $c' \in \cap_n A_n$ such that $d(c, c') \leq \epsilon$. Since $c' \in A_n$ for all $n \in \{1, 2, \ldots\}$, $c \in A_n^{\oplus \epsilon}$ for all $n \in \{1, 2, \ldots\}$. Hence, $c \in \cap_n A_n^{\oplus \epsilon}$. Therefore, $(\cap_n A_n)^{\oplus \epsilon} \subseteq \cap_n A_n^{\oplus \epsilon}$. $\square$

**Lemma A.2.** Let $A_n \in \mathcal{B}(\mathcal{X})$ for $n \in \{1, 2, \ldots\}$. Then,

$$(\cup_n A_n)^{\oplus \epsilon} = \cup_n A_n^{\oplus \epsilon}.$$

*Proof.* Suppose $a \in (\cup_n A_n)^{\oplus \epsilon}$. Then there exists $a_i \in A_i$ for some $i \in \mathbb{N}$ such that $d(a, a_i) \leq \epsilon$. Hence, $a \in A_i^{\oplus \epsilon} \subseteq \cup_n A_n^{\oplus \epsilon}$. Therefore, $(\cup_n A_n)^{\oplus \epsilon} = \cup_n A_n^{\oplus \epsilon}$.

Suppose $b \in \cup_n A_n^{\oplus \epsilon}$. Then $b \in A_j^{\oplus \epsilon}$ for some $j \in \mathbb{N}$. So there must exist $b' \in A_j$ such that $d(b, b') \leq \epsilon$. Since $b' \in \cup_n A_n$, we get that $b \in (\cup_n A_n)^{\oplus \epsilon}$. Therefore, $\cup_n A_n^{\oplus \epsilon} \subseteq (\cup_n A_n)^{\oplus \epsilon}$. $\square$

**Lemma A.3.** Let $(F_n)$ be a sequence of closed sets in $\mathcal{X}$ such that $F_k \supseteq F_{k+1}$ for $k \in \mathcal{N}$. Then,

$$(\cap_n F_n)^{\oplus \epsilon} = \cap_n F_n^{\oplus \epsilon}.$$

*Proof.* Suppose $x \in (\cap_n F_n)^{\oplus \epsilon}$. Then there exists $x' \in \cap_n F_n$ such that $d(x, x') \leq \epsilon$. Since $x' \in F_n$ for all $n \in \mathbb{N}$, $x \in F_n^{\oplus \epsilon}$ for all $n \in \mathbb{N}$. Hence, $x \in \cap_n F_n^{\oplus \epsilon}$ and therefore $(\cap_n F_n)^{\oplus \epsilon} \subseteq \cap_n F_n^{\oplus \epsilon}$. We will now show the set inclusion in the opposite direction.

Let $x \in \cap_n F_n^{\oplus \epsilon}$. Then $x \in F_n^{\oplus \epsilon}$ for all $n \in \mathbb{N}$. Hence, there exists $x_n \in F_n$ such that $d(x, x_n) \leq \epsilon$ for all $n \in \mathbb{N}$. Since $(x_n)$ is a bounded sequence, it has a subsequence $(x_{n_k})$ that converges to some $x^*$. We claim that $x^* \in F := \cap_n F_n$. Indeed, for any $m \in \mathbb{N}$, the tail of the subsequence $(x_{n_k})$ with indices greater than $m$ is contained in $F_m$. Since $F_m$ is closed, $x^*$ must be in $F_m$. Since the choice of $m$ was arbitrary, $x^* \in \cap_m F_m = F$. Hence, $x \in F^{\oplus \epsilon}$ because $d(x, x^*) \leq \epsilon$. Therefore, $\cap_n F_n^{\oplus \epsilon} \subseteq F^{\oplus \epsilon}$. $\qquad \square$

**Lemma A.4.** Let $A \in \mathcal{B}(\mathcal{X})$. Let $(\gamma_n)_{n=1}^{\infty}$ be a non-negative, monotonically decreasing sequence converging to 0. Let $\overline{A}$ denote the closure of $A$ in $\mathcal{X}$. Then, $A^{\gamma_n} \downarrow \overline{A}$.

*Proof.* We know $\overline{A} \subseteq \overline{A}^{\gamma_n} = A^{\gamma_n}$ for all $n$. Hence $\overline{A} \subseteq \lim_{n \to \infty} \bigcap_{k=1}^{n} A^{\gamma_k}$.

Suppose $x \in \lim_{n \to \infty} \bigcap_{k=1}^{n} A^{\gamma_k}$. Then it must be that $d(x, \overline{A}) = 0$ because otherwise $x$ would not lie in $A^{\gamma_n}$ for all large enough $n$. Since $d(x, \overline{A}) = 0$, we can find a sequence of points in $\overline{A}$ that tend to $x$. But since $\overline{A}$ is closed, we must have $x \in \overline{A}$. Hence, $\lim_{n \to \infty} \bigcap_{k=1}^{n} A^{\gamma_k} \subseteq \overline{A}$. $\qquad \square$

**Lemma A.5.** Let $\epsilon_1 > \epsilon_2 > 0$ and $A \in \mathcal{B}(\mathcal{X})$. Then for any $\delta \in (0, \epsilon_1 - \epsilon_2)$, $A^{\epsilon_1 - \epsilon_2 - \delta} \subseteq (A^{\epsilon_1})^{-\epsilon_2}$.

*Proof.* Recall that for $\epsilon > 0$, $A^{-\epsilon} = ((A^c)^{\epsilon})^c$. From the definition, $x \in A^{-\epsilon}$ if and only if $d(x, A^c) > \epsilon$.

Let $\delta \in (0, \epsilon_1 - \epsilon_2)$ and $x \in A^{\epsilon_1 - \epsilon_2 - \delta}$. Then, $d(x, A) \leq \epsilon_1 - \epsilon_2 - \delta$. Consider any $y \in (A^{\epsilon_1})^c$. Then, $d(y, A) > \epsilon_1$. By the reverse triangle inequality,

$$d(x, y) \geq d(y, A) - d(x, A) > \epsilon_1 - (\epsilon_1 - \epsilon_2 - \delta) = \epsilon_2 + \delta.$$

Hence,

$$d(x, (A^{\epsilon_1})^c) = \inf_{y \in (A^{\epsilon_1})^c} d(x, y) \geq \epsilon_2 + \delta > \epsilon_2.$$

Therefore, $x \in (A^{\epsilon_1})^{-\epsilon_2}$. $\qquad \square$

**Lemma A.6.** Let $A \in \mathcal{B}(\mathcal{X})$. Then, $\mathbb{1}\{x \in A^{\oplus \epsilon}\} = \sup_{x' \in B_\epsilon(x)} \mathbb{1}\{x' \in A\}$.

*Proof.* Suppose $x \in A^{\oplus \epsilon}$. Then there exists $x' \in A$ such that $x' \in B_\epsilon(x)$. Hence, $\sup_{x' \in B_\epsilon(x)} \mathbb{1}\{x' \in A\} = 1$.

Suppose $x \in \mathcal{X}$ is such that $\sup_{x' \in B_\epsilon(x)} \mathbb{1}\{x' \in A\} = 1$. Then there is a sequence $(x_n)_{n=1}^{\infty}$ such that $d(x, x_n) \leq \epsilon$ and $x_n \in A$ for all $n$. Since $(x_n)$ is a bounded sequence in a closed set, $B_\epsilon(x)$, it has a subsequence that converges to some $x^*$ such that $d(x, x^*) \leq \epsilon$ and $x^* \in A$. Hence, $x \in B_\epsilon(x^*) \subseteq A^{\oplus \epsilon}$. $\qquad \square$

# B  Proofs from Section 4

*Proof of Lemma 4.1.* We prove the above statement by using a counterexample motivated from Example 2.4 in [25]. For any $\epsilon > 0$, there exists a Borel measurable set $S \subseteq [-\epsilon, \epsilon]^2$ such that its projection onto the first coordinate is not Borel measurable ([25], Theorem 6.7.2 and Theorem 6.7.11 in [4]). That is, $S \in \mathcal{B}(\mathbb{R}^2)$ but $S_1 := \{x_1 \in \mathbb{R} : (x_1, x_2) \in S\} \notin \mathcal{B}(\mathbb{R})$.

Define a homeomorphism $\phi : \mathbb{R}^3 \to \mathbb{R}^3$ as $\phi(x_1, x_2, x_3) := (x_1, x_2, \sqrt{\epsilon^2 - x_2^2})$. $\phi$ maps the plane $[-\epsilon, \epsilon]^2 \times \{0\}$ onto the half-cylinder, $\{(x_1, x_2, x_3) \in \mathbb{R}^3 : x_1 \in [-\epsilon, \epsilon], x_2^2 + x_3^2 = \epsilon^2, x_3 \geq 0\}$, of radius $\epsilon$. Let $A := \phi(S \times \{0\})$. Then $A \in \mathcal{B}(\mathbb{R}^3)$ because $S \times \{0\} \in \mathcal{B}(\mathbb{R}^3)$. We have the following equality.

$$A^{\oplus \epsilon} \cap (\mathbb{R} \times \{0\}^2) = S_1 \times \{0\}^2$$

Suppose $A^{\oplus \epsilon} \in \mathcal{B}(\mathbb{R}^3)$. Then the above equality implies that $S_1 \in \mathcal{B}(\mathbb{R})$ contradicting our choice of $S$. Hence, $A^{\oplus \epsilon} \notin \mathcal{B}(\mathbb{R}^3)$.

$\qquad \square$

*Proof of Lemma 4.2.* Recall that an analytic set is a continuous image of a Borel set in a Polish space. Although an analytic set need not be Borel measurable, it is always universally measurable, i.e., measurable with respect to any measure defined on a complete measure space [1].

We will now show that if $A \in \mathcal{B}(\mathcal{X})$, then $A^{\oplus \epsilon}$ is an analytic set, thus showing that it is measurable in the complete measure space $(\mathcal{X}, \overline{\mathcal{B}}(\mathcal{X}))$.

Define $D = \{(x, x') \in \mathcal{X}^2 : d(x, x') \leq \epsilon\}$. $D$ is Borel measurable because it is the preimage of the Borel set $(-\infty, \epsilon]$ under the Borel measurable function $d$. Define $f : D \to \mathbb{R}$ as $f(x, x') = -\mathbb{1}\{x' \in A\}$. For $c \in \mathbb{R}$, we have the following.

$$\{(x, x') \in \mathcal{X}^2 : f(x, x') < c\} = \begin{cases} \phi & c \leq -1, \\ (\mathcal{X} \times A) \cap D & c \in (-1, 0], \\ \mathcal{X}^2 & c > 0. \end{cases}$$

Since $A \in \mathcal{B}(\mathcal{X})$ and $D \in \mathcal{B}(\mathcal{X}^2)$, $(\mathcal{X} \times A) \cap D \in \mathcal{B}(\mathcal{X}^2)$. Hence, by Definition 7.21 in [1], $f$ is a lower semianalytic function. By Proposition 7.47 in [1], the function $f^* : \mathcal{X} \to \mathbb{R}$ defined as $f^*(x) := \inf_{x' \in B_\epsilon(x)} f(x, x')$ is lower semianalytic. By Lemma A.6, we have

$$f^*(x) = \inf_{x' \in B_\epsilon(x)} -\mathbb{1}\{x' \in A\} = - \sup_{x' \in B_\epsilon(x)} \mathbb{1}\{x' \in A\} = -\mathbb{1}\{x \in A^{\oplus \epsilon}\}.$$

By Definition 7.21 in [1], it follows that $A^{\oplus \epsilon}$ is an analytic set. By Corollary 7.42.1 in [1], $A^{\oplus \epsilon} \in \overline{\mathcal{B}}(\mathcal{X})$.

$\square$

*Proof of Lemma 4.3.* Let $\beta = 1/4$. Take any $e \in E$. Since $E = A^\epsilon \backslash A^{\epsilon)}$, we have the following two implications: 1) $E \subseteq A^\epsilon$ which implies that $d(e, A) \leq \epsilon$, and 2) $E \cap A^{\epsilon)} = \varnothing$ which implies that $d(e, A) > \epsilon$. Combining the two implications, we get that $d(e, A) = \epsilon$. Hence, for every $r \in (0, \epsilon]$, there must exist an $a_r \in A$ such that $\epsilon \leq \|e - a_r\| < \epsilon + r/4$. We pick an $x' \in \mathcal{X}$ on the line segment joining $a_r$ and $x$ as follows.

$$t := \frac{r}{2\|e - a_r\|},$$
$$x' := t a_x + (1 - t) x.$$

Since $\|x - a_r\| \in [\epsilon, \epsilon + r/4)$ and $r \in (0, \epsilon]$, it is clear that $t \in (0, 1/2)$. From the definition of $x'$, it follows that $\|x' - e\| = t\|e - a_r\| = r/2$. We will now show that $B_{\beta r}(x') \subseteq B_r(e) \backslash E$. For any $y \in B_{\beta r}(x')$, we have the following.

$$\|y - e\| \leq \|y - x'\| + \|x' - e\| \leq \beta r + r/2 < r.$$

Hence, $y \in B_r(e)$. Moreover,

$$\|y - a_r\| \leq \|y - x'\| + \|x' - a_r\| \leq \beta r + (\|e - a_r\| - r/2) < \epsilon.$$

Hence, $y \in A^{\epsilon)}$ and so $y \notin E$. Therefore, $B_{\alpha r}(x') \subseteq B_r(e) \backslash E$. Hence, we have the following property (call it $(*)$): For any $e \in E$ and any $r \in (0, \epsilon]$, there is an $x' \in \mathcal{X}$ such that $B_{\beta r}(x') \subseteq B_r(e) \backslash E$.

Let $\alpha = \beta(1 - \beta)$. Take any $x \in \mathcal{X}$ and $r \in (0, \epsilon]$. We will now show that there exists $x' \in \mathcal{X}$ such that $B_{\alpha r}(x') \subseteq B_r(x) \backslash E$.

Suppose $x \in E$. Then by the property $(*)$, there exists $x' \in \mathcal{X}$ such that $B_{\alpha r}(x') \subseteq B_{\beta r}(x') \subseteq B_r(x) \backslash E$. Suppose on the other hand $x \notin E$. If $B_{\beta r}(x) \cap E = \varnothing$, then choosing $x' = x$ we have $B_{\alpha r}(x') \subseteq B_{\beta r}(x') \subseteq B_r(x) \backslash E$. If not, then there exists $e \in B_{\beta r}(x) \cap E$. We claim that $B_{(1-\beta)r}(e) \subseteq B_{\beta r}(x)$. Indeed, for any $y \in B_{(1-\beta)r}(e)$ we have

$$\|y - x\| \leq \|y - e\| + \|e - x\| \leq (1 - \beta)r + \beta r = r.$$

Since $(1 - \beta)r \in (0, \epsilon]$, by the property $(*)$, there exists $x' \in \mathcal{X}$ such that $B_{\alpha r}(x') = B_{\beta(1-\beta)r}(x') \subseteq B_{(1-\beta)r}(x) \backslash E \subseteq B_r(x) \backslash E$.

$\square$

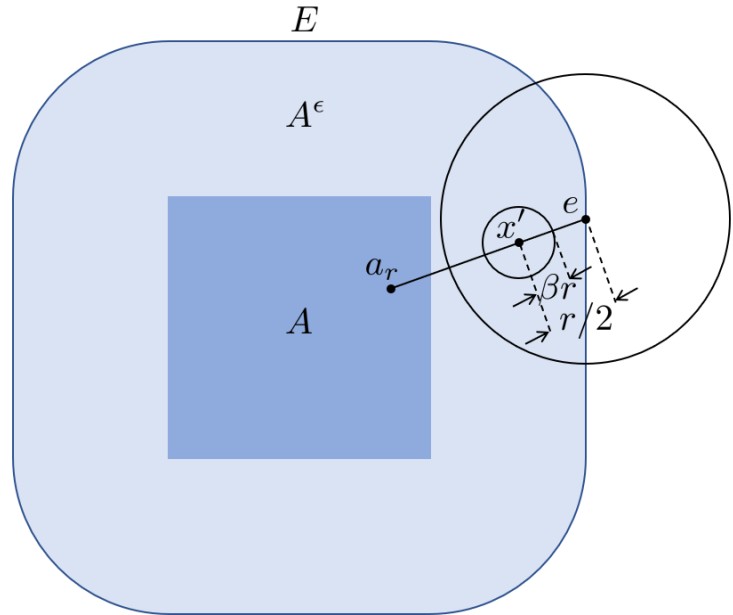

Figure 2: Choice of $x' \in \mathcal{X}$ and $\alpha \in (0, 1)$ such that $B_{\alpha r}(x') \subseteq B_r(x) \backslash E$ when $x \in E$.

*Proof of Theorem 4.2.* By Lemma 4.3 $A^\epsilon \backslash A^{\epsilon)}$ is porous, and so $\lambda(A^\epsilon \backslash A^{\epsilon)}) = 0$. Hence, $\lambda(A^\epsilon) = \lambda(A^{\epsilon)})$. Using the fact that $A^{\epsilon)} \subseteq A^{\oplus\epsilon} \subseteq A^\epsilon$, we have $A^{\oplus\epsilon} \backslash A^{\epsilon)} \subseteq A^\epsilon \backslash A^{\epsilon)}$. Hence, $\lambda(A^{\oplus\epsilon} \backslash A^{\epsilon)}) = 0$. Therefore, $A^{\oplus\epsilon} \in \mathcal{L}(\mathcal{X})$ and $\lambda(A^{\oplus\epsilon}) = \lambda(A^\epsilon) = \lambda(A^{\epsilon)})$.

Since $A^{\oplus\epsilon}, (A^c)^{\oplus\epsilon} \in \mathcal{L}(\mathcal{X})$, $R_{\oplus\epsilon}(\ell_{0/1}, A)$ is well-defined. If $p_0$ and $p_1$ are absolutely continuous with respect to the Lebesgue measure, the equation $R_{\oplus\epsilon}(\ell_{0/1}, A) = R_\epsilon(\ell_{0/1}, A)$ follows from the previous conclusion that $\lambda(A^{\oplus\epsilon}) = \lambda(A^\epsilon)$. $\qquad\square$

## C   Proofs from Section 5

### C.1   Proofs from Section 5.1

*Proof of Lemma 5.1.* Let $\mu' \in \mathcal{P}(\mathcal{X})$ be such that $W_\infty(\mu, \mu') \leq \epsilon$. Then there exists a coupling $\lambda \in \Pi(\mu', \mu)$ such that for $(x, x') \sim \lambda$, $d(x, x') \leq \epsilon$ $\lambda$-a.e. Hence,

$$\mu'(A) = \lambda(A \times \mathcal{X}) = \lambda(A \times A^{\oplus\epsilon}) \leq \lambda(\mathcal{X} \times A^{\oplus\epsilon}) = \mu(A^{\oplus\epsilon}).$$

Since the choice of $\mu'$ was arbitrary in the set $\{\nu \in \mathcal{P}(\mathcal{X}) : W_\infty(\mu, \nu) \leq \epsilon\}$, we have,

$$\sup_{W_\infty(\mu, \mu') \leq \epsilon} \mu'(A) \leq \mu(A^{\oplus\epsilon}).$$

Now we show the inequality in the opposite direction. Like in the proof of Lemma 4.2, consider the function $f : D \to \mathbb{R}$ defined as $f(x, x') = -\mathbb{1}\{x' \in A\}$, where $D = \{(x, x') \in \mathcal{X}^2 : d(x, x') \leq \epsilon\}$. Define $f^* : \mathcal{X} \to \mathbb{R}$ as $f^*(x) := \inf_{x' \in B_\epsilon(x)} f(x, x')$. As shown in the proof of Lemma 4.2, $f^*(x) = -\mathbb{1}\{x \in A^{\oplus\epsilon}\}$. By Proposition 7.50(a) in [1], there exists a universally measurable function $\phi : \mathcal{X} \to \mathcal{X}$ such that $|f^*(x) - f(x, \phi(x))| < \delta$ for any $\delta > 0$. Since $f$ and $f^*$ are both 0-1 valued functions, we get $f^*(x) = f(x, \phi(x))$ for all $x \in \mathcal{X}$ by choosing $\delta = 1/2$. Moreover, by Proposition 7.50(a) in [1], $Gr(\phi) \subseteq D$ i.e., $d(x, \phi(x)) \leq \epsilon$ for all $x \in \mathcal{X}$. Therefore,

$$\sup_{W_\infty(\mu, \mu') \leq \epsilon} \mu'(A) \geq \phi_\sharp \mu(A) = \mu(\phi^{-1}(A)) = \mu(A^{\oplus\epsilon}).$$

Hence, $\sup_{W_\infty(\mu, \mu') \leq \epsilon} \mu'(A) = \phi_\sharp \mu(A) = \mu(A^{\oplus\epsilon})$ for any set $A \in \mathcal{B}(\mathcal{X})$. $\qquad\square$

*Proof of Theorem 5.1.* Since $A \in \mathcal{B}(\mathcal{X})$, $A^c \in \mathcal{B}(\mathcal{X})$ and by Lemma 4.2, $A^{\oplus\epsilon}, (A^c)^{\oplus\epsilon} \in \overline{\mathcal{B}}(\mathcal{X})$. Therefore $R_{\oplus\epsilon}(\ell_{0/1}, A)$ is well-defined.

By Lemma 5.1, we have

$$
\begin{aligned}
R_{\Gamma_\epsilon}(\ell_{0/1}, A) &= \sup_{\substack{W_\infty(p_0, p_0') \le \epsilon \\ W_\infty(p_1, p_1') \le \epsilon}} \frac{T}{T+1} p_0'(A) + \frac{1}{T+1} p_1'((A^c)) \\
&= \frac{T}{T+1}\left(\sup_{W_\infty(p_0, p_0') \le \epsilon} p_0'(A)\right) + \frac{1}{T+1}\left(\sup_{W_\infty(p_1, p_1') \le \epsilon} p_1'((A^c))\right) \\
&= \frac{T}{T+1} p_0(A^{\oplus\epsilon}) + \frac{1}{T+1} p_1((A^c)^{\oplus\epsilon}) \\
&= R_{\oplus\epsilon}(\ell_{0/1}, A)
\end{aligned}
$$

By Lemma 5.1 again, the supremum over $p_0'$ and $p_1'$ in $R_{\Gamma_\epsilon}(\ell_{0/1}, A)$ is attained by measures pushed forward from $p_0$ and $p_1$ via some measurable maps $f_0$ and $f_1$. From this, the remaining assertions of the theorem follow. $\qquad\square$

## C.2 Proofs from Section 5.2

*Proof of Lemma 5.2.* The following properties of $v$ are trivially true: $v(\phi) = 0$, $v(\mathcal{X}) = 1$ and $v(A) \le v(B)$ for $A \subseteq B$.

Consider a sequence of sets $(A_n)$ in $\mathcal{X}$ such that $A_k \subseteq A_{k+1}$ for $k \in \mathcal{N}$. Let $A = \cup_n A_n$. That is, $A_n \uparrow A$. Then by Lemma A.1 we have, $A^{\oplus\epsilon} = \cup_n A_n^{\oplus\epsilon}$. Hence, $A_n^{\oplus\epsilon} \uparrow A^{\oplus\epsilon}$ and by the continuity of measure, $v(A_n) = \mu(A_n^{\oplus\epsilon}) \uparrow \mu(A^{\oplus\epsilon}) = v(A)$.

Consider a sequence of closed sets $(F_n)$ in $\mathcal{X}$ such that $F_k \supseteq F_{k+1}$ for $k \in \mathbb{N}$. Let $F = \cap_n F_n$. That is, $F_n \downarrow F$. By Lemma A.3, $F_n^{\oplus\epsilon} \downarrow F^{\oplus\epsilon}$. Hence, by the continuity of measure, we have $v(F_n) = \mu(F_n^{\oplus\epsilon}) \downarrow \mu(F^{\oplus\epsilon}) = v(F)$.

For any two sets $A, B \in \mathcal{L}(\mathcal{X})$,

$$
\begin{aligned}
v(A \cup B) &= \mu((A \cup B)^{\oplus\epsilon}) \\
&\overset{(i)}{=} \mu(A^{\oplus\epsilon} \cup B^{\oplus\epsilon}) \\
&= \mu(A^{\oplus\epsilon}) + \mu(B^{\oplus\epsilon}) - \mu(A^{\oplus\epsilon} \cap B^{\oplus\epsilon}) \\
&\overset{(ii)}{\le} \mu(A^{\oplus\epsilon}) + \mu(B^{\oplus\epsilon}) - \mu((A \cap B)^{\oplus\epsilon}) \\
&= v(A) + v(B) - v(A \cap B),
\end{aligned}
$$

where $(i)$ follows from Lemma A.1 and $(ii)$ follows from Lemma A.2. Hence, $v$ is a 2-alternating capacity. $\qquad\square$

*Proof of Lemma 5.3.* Let $\mu' \in \mathcal{P}(\mathcal{X})$ be such that $W_\infty(\mu, \mu') \le \epsilon$. Then there exists a coupling $\gamma \in \Pi(\mu', \mu)$ such that for $(x, x') \sim \gamma$, $d(x, x') \le \epsilon$ $\gamma$-a.e. Hence,

$$
\mu'(A) = \gamma(A \times \mathcal{X}) = \gamma(A \times A^{\oplus\epsilon}) \le \gamma(\mathcal{X} \times A^{\oplus\epsilon}) = \mu(A^{\oplus\epsilon}).
$$

Since the choice of $\mu'$ was arbitrary in the set $\{\nu \in \mathcal{P}(\mathcal{X}) : W_\infty(\mu, \nu) \le \epsilon\}$, we have,

$$
\sup_{W_\infty(\mu, \mu') \le \epsilon} \mu'(A) \le \mu(A^{\oplus\epsilon}).
$$

We will now show the inequality in the reverse direction. By Lemma 5.2, $A \mapsto \mu(A^{\oplus\epsilon})$ is a 2-alternating capacity. Hence by Lemma 2.5 in [21], for any Lebesgue measurable $A \subseteq \mathcal{X}$, there exists a $\nu \in \mathcal{P}(\mathcal{X})$ such that $\nu(A) = \mu(A^{\oplus\epsilon})$ and $\nu(B) \le \mu(B^{\oplus\epsilon})$ for all Lebesgue measurable $B \subseteq \mathcal{X}$. For such a $\nu$, it is clear that $W_\infty(\mu, \nu) \le \epsilon$. Hence,

$$
\sup_{W_\infty(\mu, \mu') \le \epsilon} \mu'(A) \ge \nu(A) = \mu(A^{\oplus\epsilon}).
$$

Hence, $\sup_{W_\infty(\mu, \mu') \le \epsilon} \mu'(A) = \nu(A) = \mu(A^{\oplus\epsilon})$. $\qquad\square$

*Proof of Theorem 5.2.*

$$R_{\Gamma_\epsilon}(\ell_{0/1}, A) = \frac{T}{T+1}\left(\sup_{W_\infty(p_0,p_0')\leq\epsilon} p_0'(A)\right) + \frac{1}{T+1}\left(\sup_{W_\infty(p_1,p_1')\leq\epsilon} p_1'((A^c))\right)$$

$$\overset{(*)}{=} \frac{T}{T+1}p_0(A^{\oplus\epsilon}) + \frac{1}{T+1}p_1((A^c)^{\oplus\epsilon})$$

$$= R_{\oplus\epsilon}(\ell_{0/1}, A),$$

where $(*)$ follows from Lemma 5.3. By Lemma 5.3 again, the supremum over $p_0'$ and $p_1'$ in $R_{\Gamma_\epsilon}(\ell_{0/1}, A)$ is attained. $\qquad\square$

## D  Proofs from Section 6

We first prove a discrete version of Theorem 6.1 on a finite space.

**Lemma D.1.** Let $\mathcal{X}_n = \{x_1, \ldots, x_n\} \subseteq \mathcal{X}$. Let $\boldsymbol{p} = (p_i)_{i=1}^n, \boldsymbol{q} = (q_i)_{i=1}^n$ be such that $p_i, q_i \geq 0$ for $i \in [n]$ and $\sum_i p_i \leq \sum_i q_i$. Let $\epsilon > 0$. For $A \subseteq \mathcal{X}_n$, let $A^\epsilon := \{x \in \mathcal{X}_n : d(x, x') \leq \epsilon,$ for some $x' \in A\}$. For $A \subseteq \mathcal{X}_n$, let $\boldsymbol{p}(A) = \sum_{i:x_i\in A} p_i$ and $\boldsymbol{q}(A) = \sum_{i:x_i \in A} q_i$. For $i, j \in [n]$, let $c_{ij} = \mathbb{1}\{d(x_i, x_j) > 2\epsilon\}$. Then,

$$\max_{A\subseteq\mathcal{X}_n} \boldsymbol{p}(A) - \boldsymbol{q}(A^{2\epsilon}) = \min_{\substack{x_{ij}\geq 0 \\ \sum_j x_{ij}=p_i \\ \sum_i x_{ij}\leq q_j}} \sum_{i,j} c_{ij}x_{ij}. \tag{16}$$

*Proof.* For $i, j \in [n]$, define $d_{ij} := 1 - c_{ij}$. Then,

$$\min_{\substack{x_{ij}\geq 0 \\ \sum_j x_{ij}=p_i \\ \sum_i x_{ij}\leq q_j}} \sum_{i,j} c_{ij}x_{ij} = \sum_i p_i - \max_{\substack{x_{ij}\geq 0 \\ \sum_j x_{ij}=p_i \\ \sum_i x_{ij}\leq q_j}} \sum_{i,j} d_{ij}x_{ij} \tag{17}$$

Consider the following modification to the linear program on the right hand side of (17), where the constraint $\sum_j x_{ij} = p_i$ is replaced by $\sum_j x_{ij} \leq p_i$.

$$\max_{\substack{x_{ij}\geq 0 \\ \sum_j x_{ij}\leq p_i \\ \sum_i x_{ij}\leq q_j}} \sum_{i,j} d_{ij}x_{ij}. \tag{18}$$

We will show that the above linear program is equivalent to the linear program on the right hand side of (17). Since the above linear program is bounded and feasible, it admits a solution. Let $\{x_{ij}^*\}_{i,j\in[n]}$ be the solution to (18). Suppose there exists $m \in [n]$ such that $\sum_j x_{mj}^* < p_m$. Let $s = p_m - \left(\sum_j x_{mj}^*\right) > 0$. For $j \in [n]$, define $s_j := q_j - \sum_i x_{ij}^*$. Then,

$$\sum_j s_j = \sum_j q_j - \sum_{i,j} x_{ij}$$

$$\geq \sum_i p_i - \left(\left(\sum_{i\neq m} p_i\right) + p_m - s\right)$$

$$= s.$$

Therefore, $\sum_j s_j \geq s$. Let $k$ be the largest integer for which $\sum_{j=1}^k s_j < s$. Define,

$$y_{ij} = \begin{cases} x_{ij}^* & i \neq m, \\ x_{mj}^* + s_j & i = m, j \leq k, \\ x_{mk}^* + s - \sum_{j=1}^k s_j & i = m, j = k+1, \\ x_{mj}^* & i = m, j \geq k+1. \end{cases} \tag{19}$$

By the above definition we have,

$$\sum_j y_{ij} = \begin{cases} \sum_j x_{ij}^* & i \neq m, \\ \sum_j x_{ij}^* + s & i = m. \end{cases}$$

$$\sum_i y_{ij} = \begin{cases} \sum_i x_{mj}^* + s_j & j \leq k, \\ \sum_i x_{mk}^* + s - \sum_{j=1}^k s_j & j = k+1, \\ \sum_i x_{mj}^* & j \geq k+1. \end{cases}$$

Combining the above with the definitions of $k, s$ and $\{s_j\}_{j \in [n]}$, we see that $\sum_j y_{ij} \leq p_i$ and $\sum_i y_{ij} \leq q_j$. Moreover, $y_{ij} \geq x_{ij}$ for all $i, j \in [n]$. Hence, $\sum_{ij} d_{ij} y_{ij} \geq \sum_{ij} d_{ij} x_{ij}$. Therefore, any solution $\{x_{ij}^*\}_{i,j \in [n]}$ for which there exists $m \in [n]$ such that $\sum_j x_{mj}^* < p_m$, can be improved to a solution $\{y_{ij}\}_{i,j \in [n]}$ for which $\sum_j y_{mj} = p_m$. Hence,

$$\max_{\substack{x_{ij} \geq 0 \\ \sum_j x_{ij} = p_i \\ \sum_i x_{ij} \leq q_j}} \sum_{i,j} d_{ij} x_{ij} = \max_{\substack{x_{ij} \geq 0 \\ \sum_j x_{ij} \leq p_i \\ \sum_i x_{ij} \leq q_j}} \sum_{i,j} d_{ij} x_{ij}. \tag{20}$$

Since the maximization in (20) is a linear program in canonical form, we employ the strong duality theorem (for a reference, see Chapter 6 in [29]) to get the following.

$$\max_{\substack{x_{ij} \geq 0 \\ \sum_j x_{ij} \leq p_i \\ \sum_i x_{ij} \leq q_j}} \sum_{i,j} d_{ij} x_{ij} = \min_{\substack{u_i, v_i \geq 0 \\ u_i + v_j \geq d_{ij}}} \sum_i (p_i u_i + q_i v_i). \tag{21}$$

Since $d_{ij} \in \{0, 1\}$, we may assume $u_i, v_i \leq 1$ for the minimization in (21) without violating other constraints because any decrease of $u_i, v_i$ down to 1 will only decrease the value of $\sum_i (p_i u_i + q_i v_i)$, which we seek to minimize. Defining $w_i := 1 - u_i$, we have the following from (17) and (21).

$$\min_{\substack{x_{ij} \geq 0 \\ \sum_j x_{ij} = p_i \\ \sum_i x_{ij} \leq q_j}} \sum_{i,j} c_{ij} x_{ij} = \max_{\substack{w_i, v_i \in [0,1] \\ w_i - v_j \leq c_{ij}}} \sum_i (p_i w_i - q_i v_i). \tag{22}$$

The optimal $w_i^*, v_i^*$ that achieve the maximum in (22) must lie at one of the vertices of the polyhedron supported by the hyperplanes, $w_i = 0, w_i = 1, v_i = 0, v_i = 1$ and $w_i - v_j = c_{ij}$. Hence, $w_i^*, v_i^* \in \{0, 1\}$. Moreover if $c_{ij} = 0$ and $w_i^* = 1$ for some $i, j \in [n]$, then $v_j^* = 1$. On the other hand if $c_{ij} = 1$, then $v_j^*$ can be set to 0 without violating other constraints and without decreasing the maximization objective. Therefore, setting $A := \{x_i \in \mathcal{X}_n : w_i^* = 1\}$, we see that the maximum in (22) equals the maximum in (16). □

*Proof of Theorem 6.1.* Let $(\gamma_n)_{n=1}^\infty$ be a non-negative, monotonically decreasing sequence converging to 0. Let $(x_n)_{n=1}^\infty$ be a dense sequence in $\mathcal{X}$. Define a function $f : \mathcal{X} \to \{x_n\}_{n=1}^\infty$ such that $f(x) = x_k$ for the least integer $k$ with $d(x, x_k) < \gamma_n$. Let $H_n = \{x_1, \ldots, x_n\}$. Let $s_n$ be the least positive integer such that,

$$\mu(f^{-1}(H_{s_n - 1})) > \mu(\mathcal{X}) - \gamma_n, \tag{23}$$
$$\nu(f^{-1}(H_{s_n - 1})) > \nu(\mathcal{X}) - \gamma_n. \tag{24}$$

Given $n$, construct a discrete measure $\mu_n$ supported on the finite set $H_{s_n}$ such that $\mu_n(x_k) := \mu(f^{-1}(x_k))$ for $k \in [s_n - 1]$ and $\mu_n(\mathcal{X}) = \mu(\mathcal{X})$. Similarly, construct $\nu_n$ supported on $H_{s_n}$ such that $\nu_n(x_k) := \nu(f_n^{-1}(x_k))$ for $k \in [s_n - 1]$ and $\nu_n(\mathcal{X}) = \nu(\mathcal{X})$.

Let $A \in \mathcal{B}(\mathcal{X})$. We have,

$$\mu_n(A) \overset{(i)}{=} \mu_n(A \cap H_{s_n})$$
$$\overset{(ii)}{<} \mu_n(A \cap H_{s_n - 1}) + \gamma_n$$
$$\overset{(iii)}{=} \mu(f^{-1}(A \cap H_{s_n - 1})) + \gamma_n$$
$$\overset{(iv)}{\leq} \mu(A^{\gamma_n}) + \gamma_n, \tag{25}$$

where $(i)$ follows from the fact that $\mu_n$ is supported on $H_{s_n}$, $(ii)$ follows from (23), $(iii)$ follows from the definition of $\mu_n$ and $(iv)$ follows because of the following: For any $y \in A \cap H_{s_n-1}$, $f^{-1}(y) \subseteq \{x \in \mathcal{X} : d(x,y) < \gamma_n\} \subseteq A^{\gamma_n}$. Hence, $f^{-1}(A \cap H_{s_n-1}) \subseteq A^{\gamma_n}$. Applying (25), with $A^c$ instead of $A$, we have the following.

$$\mu(A^{-\gamma_n}) - \gamma_n \leq \mu_n(A) \leq \mu(A^{\gamma_n}) + \gamma_n. \tag{26}$$

Letting $n \to \infty$ in (26) and using Lemma A.4, we get that $\limsup_n \mu_n(A) \leq \mu(A)$ for all closed subsets $A$ of $\mathcal{X}$. Hence, by applying the Portmanteau theorem (Theorem 2.1 in [3]), we conclude that the sequence of measures $(\mu_n)_{n=1}^\infty$ converges weakly to $\mu$. Similarly, $\nu_n \to \nu$ weakly.

For any fixed $n$, we apply Lemma D.1 to the measures $\mu_n, \nu_n$ on the finite space $H_{s_n}$ to get the following.

$$\max_{A \subseteq H_{s_n}} \mu_n(A) - \nu_n(A^{2\epsilon+4\gamma_n}) = \min_{\substack{x_{ij} \geq 0 \\ \sum_j x_{ij}=\mu_n(x_i) \\ \sum_i x_{ij} \leq \nu_n(x_j)}} \sum_{i,j} x_{ij} \mathbb{1}\{d(x_i, x_j) > 2\epsilon + 4\gamma_n\}, \tag{27}$$

where the indices $i, j$ run over $[s_n]$. We have that $\mu_n(\mathcal{X}) = \mu(\mathcal{X}) \leq \nu(\mathcal{X}) = \nu(\mathcal{X})$. Define a coupling $\pi_n \in \Pi(\mu_n, \nu_n)$ supported on $H_{s_n} \times H_{s_n}$ using the optimal solution $\{x_{ij}\}_{i,j \in [s_n]}$ to the minimization in (27) by setting $\pi_n(i,j) = x_{ij}^*$. Let $T_n \subseteq H_{s_n}$ be the set that achieves the maximum in (27).

We will now construct a candidate coupling for the infimum in (8). Since $\mu, \nu$ are finite measures on a Polish space, they are tight (see for example, Theorem 1.3 in [3]). Hence, given a $\delta > 0$, there exists a compact set $K \subseteq \mathcal{X}$ such that $\min\{\mu(K^c), \nu(K^c)\} < \delta/3$. Since $\mu_n$ and $\nu_n$ converge weakly to $\mu$ and $\nu$ respectively, choose $N$ large enough so that $\min\{\mu_n(K^c), \nu_n(K^c)\} < \delta/2$ for all $n \geq N$. Let $\nu_n'$ be the second marginal of the coupling $\pi_n$. Then, $\nu_n' \preceq \nu_n$. By union bound, we have the following.

$$\pi_n((K \times K)^c) \leq \mu_n(K^c) + \nu_n'(K^c) \leq \mu_n(K^c) + \nu_n(K^c) < \delta. \tag{28}$$

Hence, the sequence $(\pi_n)_{n \geq N}$ is uniformly tight. Hence, by Prokhorov's theorem (for reference, see Theorem 5.1 in [3]), there is a subsequence $(\pi_{n_k})$ of $(\pi_n)_{n \geq N}$ that converges weakly to some measure $\pi^* \in \mathcal{M}(\mathcal{X} \times \mathcal{X})$. Moreover, $\pi^* \in \Pi(\mu, \nu)$ by virtue of the constraints imposed on the converging subsequence of $(\pi_n)_{n \geq N}$.

Let $\Phi = \sup_{A \in \mathcal{B}(\mathcal{X})} \mu(A) - \nu(A^{2\epsilon})$ and $\Psi = \mathcal{T}_{c_\epsilon}(\mu, \nu)$. For any $n$ we have,

$$\begin{aligned}
\pi_n(d(x_i, x_j) > 2\epsilon + 4\gamma_n) &\overset{(i)}{=} \mu_n(T_n) - \nu_n(T_n^{2\epsilon+4\gamma_n}) \\
&\overset{(ii)}{\leq} (\mu(T_n^{\gamma_n}) + \gamma_n) - (\nu((T_n^{2\epsilon+4\gamma_n})^{-\gamma_n}) - \gamma_n) \\
&\overset{(iii)}{\leq} \mu(T_n^{\gamma_n}) - \nu((T_n^{2\epsilon+4\gamma_n-\gamma_n-\gamma_n/2}) + 2\gamma_n \\
&\leq \mu(T_n^{2\gamma_n}) - \nu((T_n^{2\epsilon+2\gamma_n}) + 2\gamma_n \\
&\overset{(iv)}{\leq} \Phi + 2\gamma_n, \tag{29}
\end{aligned}$$

where $(i)$ follows from the definition of $\pi_n$ and $T_n$, $(ii)$ follows from (26), $(iii)$ follows from Lemma A.5 and $(iv)$ follows from the definition of $\Phi$. Further,

$$\begin{aligned}
\Psi &= \inf_{\pi \in \Pi(\mu, \nu)} \pi[d(x, x') > 2\epsilon] \\
&\overset{(i)}{\leq} \pi^*[d(x, x') > 2\epsilon] \\
&\overset{(ii)}{\leq} \liminf_{n_k} \pi_{n_k}[d(x, x') > 2\epsilon] \\
&\leq \limsup_n \pi_n[d(x, x') \geq 2\epsilon] \\
&\overset{(iii)}{\leq} \Phi, \tag{30}
\end{aligned}$$

where $(i)$ follows because $\pi^* \in \Pi(\mu, \nu)$, $(ii)$ follows from Portmanteau's theorem because $(\pi_{n_k})$ that converges to $\pi^*$ and the set $\{(x, x') \in \mathcal{X}^2 : d(x, x') > 2\epsilon\}$ is an open set, and $(iii)$ follows by taking $n \to \infty$ in (29).

To show the inequality $\Phi \leq \Psi$, consider a sequence of measures $(\lambda_n)_{n=1}^\infty$ such that $\lambda_n \in \Pi(\mu, \nu)$ and $\lim_n \lambda_n[d(x, x') > \epsilon] = \Psi$. For any $A \in \mathcal{B}(\mathcal{X})$,

$$\mu(A) = \lambda_n[x \in A, x' \in A^\epsilon] + \lambda_n[x \in A, x' \notin A^\epsilon]$$
$$\leq \nu(A^\epsilon) + \lambda_n[d(x, x') > \epsilon].$$

Letting $n \to \infty$, we have $\mu(A) - \nu(A^\epsilon) \leq \Psi$ for all $A \in \mathcal{B}(\mathcal{X})$. Hence, $\Phi \leq \Psi$. Combining this with (30), we conclude $\Phi = \Psi$. $\qquad\square$

*Proof of Lemma 6.1.* We have,

$$\sup_{A \in \mathcal{B}(\mathcal{X})} \mu(A) - \nu(A^{2\epsilon}) \overset{(i)}{=} \sup_{A \text{ closed}} \mu(A) - \nu(A^{2\epsilon})$$

$$\overset{(ii)}{\leq} \sup_{A \text{ closed}} \mu((A^{\oplus\epsilon})^{\ominus\epsilon}) - \mu((A^{\oplus\epsilon})^{\oplus\epsilon})$$

$$\overset{(iii)}{\leq} \sup_{A \in \mathcal{B}(\mathcal{X})} \mu(A^{\ominus\epsilon}) - \nu(A^{\oplus\epsilon}),$$

where $(i)$ follows because we may assume that the supremum of $\mu(A) - \nu(A^{2\epsilon})$ is achieved by a closed set. Indeed, $\mu(\overline{A}) - \nu(\overline{A}^{2\epsilon}) \geq \mu(A) - \nu(A^{2\epsilon})$ because $\overline{A} \supseteq A$ and $\overline{A}^{2\epsilon} = A^{2\epsilon}$. $(ii)$ follows from the following two facts: 1) $A \subseteq (A^{\oplus\epsilon})^{\ominus\epsilon}$ (see Lemma 3.3 in [35]), and 2) $A^\epsilon = A^{\oplus\epsilon}$ for closed sets $A$ (see Lemma 3.2 in [35]). $(iii)$ follows from Lemma 4.2 because $\mu, \nu \in \overline{\mathcal{P}}(\mathcal{X})$ and $A^{\ominus\epsilon}, A^{\oplus\epsilon} \in \overline{\mathcal{B}}(\mathcal{X})$ whenever $A \in \mathcal{B}(\mathcal{X})$.

Now, we show that the above inequality also holds in the opposite direction. Let $\mu' = \mu/t \in \overline{\mathcal{P}}(\mathcal{X})$ for some fixed $t > 0$. For $x, y \in \mathcal{X}$, define the cost function $c(x, y) = \mathbb{1}\{d(x, y) > 2\epsilon\}$. For any $\nu' \in \overline{\mathcal{P}}(\mathcal{X})$, we have the following from Kantorovich duality theorem.

$$D_\epsilon(\mu', \nu') = \sup_{\phi(x) + \psi(y) \leq c(x, y)} \int \phi d\mu' + \int \psi d\nu',$$

For any $A \in \mathcal{B}(\mathcal{X})$, define $\phi'(x) = \mathbb{1}\{x \in A^{\ominus\epsilon}\}$ and $\psi'(y) = -\mathbb{1}\{y \in A^{\oplus\epsilon}\}$. We will now show that $\phi'(x) + \psi'(y) \leq c(x, y)$. If $x, y$ are such that $c(x, y) = 1$, the inequality holds trivially. Suppose on the other hand, $x, y$ are such that $c(x, y) = 0$. Then $d(x, y) \leq 2\epsilon$. Hence, for any $x \in A^{\ominus\epsilon}$, we have $y \in (A^{\ominus\epsilon})^{\oplus 2\epsilon} = ((A^{\ominus\epsilon})^{\oplus\epsilon})^{\oplus\epsilon} \subseteq A^{\oplus\epsilon}$ (the set inclusion here follows from Lemma 3.3 in [35]). Therefore,

$$\phi'(x) + \psi'(y) = \mathbb{1}\{x \in A^{\ominus\epsilon}\} - \mathbb{1}\{y \in A^{\oplus\epsilon}\} = 0 = c(x, y).$$

Hence,

$$D_\epsilon(\mu', \nu') \geq \int \phi' d\mu' + \int \psi' d\nu' = \mu'(A^{\ominus\epsilon}) - \nu'(A^{\oplus\epsilon}).$$

Now,

$$\sup_{A \in \mathcal{B}(\mathcal{X})} \mu(A) - \nu(A^{2\epsilon}) \overset{(*)}{=} t \inf_{\substack{\nu' \in \overline{\mathcal{P}}(\mathcal{X}) \\ \nu' \preceq \nu/t}} D_\epsilon(\mu', \nu')$$

$$\geq t \inf_{\substack{\nu' \in \overline{\mathcal{P}}(\mathcal{X}) \\ \nu' \preceq \nu/t}} \mu'(A^{\ominus\epsilon}) - \nu'(A^{\oplus\epsilon})$$

$$= t\mu'(A^{\ominus\epsilon}) - t \sup_{\substack{\nu' \in \overline{\mathcal{P}}(\mathcal{X}) \\ \nu' \preceq \nu/t}} \nu'(A^{\oplus\epsilon})$$

$$\geq \mu(A^{\ominus\epsilon}) - t\frac{\nu}{t}(A^{\oplus\epsilon})$$

$$= \mu(A^{\ominus\epsilon}) - \nu(A^{\oplus\epsilon}),$$

where $(*)$ follows from Theorem 6.1. Since the above inequality is valid for any $A \in \mathcal{B}(\mathcal{X})$, we get the following.

$$\sup_{A \in \mathcal{B}(\mathcal{X})} \mu(A) - \nu(A^{2\epsilon}) \geq \sup_{A \in \mathcal{B}(\mathcal{X})} \mu(A^{\ominus \epsilon}) - \nu(A^{\oplus \epsilon}).$$

$\square$

*Proof of Theorem 6.2.*

$$\inf_{A \in \mathcal{B}(\mathcal{X})} R_{\oplus \epsilon}(\ell_{0/1}, A) = \inf_{A \in \mathcal{B}(\mathcal{X})} \frac{1}{T+1} \left[ Tp_0(A^{\oplus \epsilon}) + p_1((A^c)^{\oplus \epsilon}) \right]$$

$$= \frac{1}{T+1} \left[ 1 - \sup_{A \in \mathcal{B}(\mathcal{X})} \left( p_1(A^{\ominus \epsilon}) - Tp_0(A^{\oplus \epsilon}) \right) \right]$$

$$\stackrel{(i)}{=} \frac{1}{T+1} \left[ 1 - \sup_{A \in \mathcal{B}(\mathcal{X})} \left( p_1(A) - Tp_0(A^{2\epsilon}) \right) \right]$$

$$\stackrel{(ii)}{=} \frac{1}{T+1} \left[ 1 - \inf_{\substack{q \in \mathcal{P}(\mathcal{X}): \\ q \preceq Tp_0}} D_\epsilon(q, p_1) \right],$$

where $(i)$ follows from Lemma 6.1 and $(ii)$ follows from Theorem 6.1. $\square$

# E    Proofs from Section 7

*Proof of Theorem 7.1.* By Lemma 5.2, the set-valued maps $A \mapsto p_0(A^{\oplus \epsilon})$ and $A^c \mapsto p_1((A^c)^{\oplus \epsilon})$ are 2-alternating capacities. Hence, the existence of $A^* \in \mathcal{L}(\mathcal{X})$ that attains the infimum on the right in (12) follows from Lemma 3.1 in [21] and the equality $R_{\oplus \epsilon}(\ell_{0/1}, A) = R_{\Gamma_\epsilon}(\ell_{0/1}, A)$ proved in Theorem 5.2. By Theorem 4.1 in [21], there exist $q_0, q_1 \in \mathcal{P}(\mathcal{X})$ such that $W_\infty(p_i, q_i) \leq \epsilon$ for $i = 0, 1$ and,

$$\inf_{A \in \mathcal{L}(\mathcal{X})} \sup_{W_\infty(p_0, p_0'), W_\infty(p_1, p_1') \leq \epsilon} r(A, p_0, p_1) = \inf_{A \in \mathcal{L}(\mathcal{X})} r(A, q_0, q_1).$$

Hence,

$$\inf_{A \in \mathcal{L}(\mathcal{X})} \sup_{W_\infty(p_0, p_0'), W_\infty(p_1, p_1') \leq \epsilon} r(A, p_0, p_1) = \inf_{A \in \mathcal{A}} r(A, q_0, q_1)$$

$$\leq \sup_{W_\infty(p_0, p_0'), W_\infty(p_1, p_1') \leq \epsilon} \inf_{A \in \mathcal{L}(\mathcal{X})} r(A, p_0, p_1).$$

The desired result follows from combining the above inequality with the max-min inequality (11). Clearly, $q_0 = p_0^*$ and $q_1 = p_1^*$. $\square$

**Lemma E.1** (Max-min Inequality). *Let $p_0, p_1 \in \mathcal{P}(\mathcal{X})$ and let $\epsilon \geq 0$. For $T > 0$, define $r : \mathcal{B}(\mathcal{X}) \times \mathcal{P}(\mathcal{X}) \times \mathcal{P}(\mathcal{X}) \to [0, 1]$ as in (10). Then,*

$$\sup_{\substack{W_\infty(p_0, p_0') \leq \epsilon \\ W_\infty(p_1, p_1') \leq \epsilon}} \inf_{A \in \mathcal{B}(\mathcal{X})} r(A, p_0', p_1') \leq \inf_{A \in \mathcal{B}(\mathcal{X})} \sup_{\substack{W_\infty(p_0, p_0') \leq \epsilon \\ W_\infty(p_1, p_1') \leq \epsilon}} r(A, p_0', p_1'). \tag{31}$$

*Proof.* For any $A \in \mathcal{B}(\mathcal{X})$ and $p_0', p_1'$ such that $W_\infty(p_i, p_i') \leq \epsilon$ $(i = 0, 1)$, we have

$$\inf_{A \in \mathcal{B}(\mathcal{X})} r(A, p_0', p_1') \leq r(A, p_0', p_1').$$

Taking supremum over $p_0'$ and $p_1'$ such that $W_\infty(p_i, p_i') \leq \epsilon$ for $i \in \{0, 1\}$ on both sides of the above inequality, we get the following for any $A \in \mathcal{B}(\mathcal{X})$.

$$\sup_{W_\infty(p_0, p_0'), W_\infty(p_1, p_1') \leq \epsilon} \inf_{A \in \mathcal{B}(\mathcal{X})} r(A, p_0', p_1') \leq \sup_{W_\infty(p_0, p_0'), W_\infty(p_1, p_1') \leq \epsilon} r(A, p_0', p_1').$$

Since the above inequality holds for any $A \in \mathcal{B}(\mathcal{X})$, we have,

$$\sup_{W_\infty(p_0, p_0'), W_\infty(p_1, p_1') \leq \epsilon} \inf_{A \in \mathcal{B}(\mathcal{X})} r(A, p_0', p_1') \leq \inf_{A \in \mathcal{B}(\mathcal{X})} \sup_{W_\infty(p_0, p_0'), W_\infty(p_1, p_1') \leq \epsilon} r(A, p_0', p_1').$$

$\square$

*Proof of Theorem 7.2.* Consider any $\mu'$ and $\nu'$ such that $W_\infty(\mu, \mu') \le \epsilon$ and $W_\infty(\nu, \nu') \le \epsilon$. Then there exist $\gamma_\mu \in \Pi(\mu, \mu')$ and $\gamma_\nu \in \Pi(\nu, \nu')$ such that

$$\mathbb{P}_{(x,x') \sim \gamma_\mu}(d(x, x') > \epsilon) = 0,$$
$$\mathbb{P}_{(x,x') \sim \gamma_\nu}(d(x, x') > \epsilon) = 0.$$

Let $\gamma' \in \Pi(\mu', \nu')$ be the coupling that achieves the optimal transport cost $D_{TV}(\mu', \nu')$. Construct a coupling $\gamma_0 \in \Pi(\mu, \nu)$ as $\gamma_0 = \gamma_\mu \circ \gamma' \circ \gamma_\nu$. Then,

$$\begin{aligned} D_\epsilon(\mu, \nu) &\le \int_{\mathcal{X}^2} \mathbb{1}\{d(x, x') > 2\epsilon\} d\gamma_0 \\ &\le \int_{\mathcal{X}^2} \mathbb{1}\{d(x, x') > 0\} d\gamma' \\ &= D_{TV}(\mu', \nu'). \end{aligned}$$

Since the above inequalty is true for any $\mu'$ and $\nu'$ such that $W_\infty(\mu, \mu') \le \epsilon$ and $W_\infty(\nu, \nu') \le \epsilon$, we have the following inequality.

$$D_\epsilon(\mu, \nu) \le \inf_{W_\infty(p_0, p_0'), W_\infty(p_1, p_1') \le \epsilon} D_{TV}(\mu', \nu').$$

Now we will show the above inequality in the reverse direction. Let $\gamma \in \Pi(\mu, \nu)$ be the coupling that achieves the optimal transport cost for $D_\epsilon(\mu, \nu)$. Let $M : \mathcal{X}^2 \to \mathcal{X}$ be a measurable midpoint map. (See [13] for why such a map exists.) That is, for all $(x, x') \in \mathcal{X}^2$ we have

$$d(x, M(x, x')) = d(x', M(x, x')) = \frac{1}{2} d(x, x').$$

Consider a transport map $T : \mathcal{X}^2 \to \mathcal{X}^2$ defined as

$$T(x, x') = \begin{cases} (M(x, x'), M(x, x')) & d(x, x') \le 2\epsilon, \\ (x, x') & \text{otherwise.} \end{cases}$$

$T$ is measurable because it is piece-wise measurable on measurable sets. Further, it follows from the definition of $M$ that each coordinate of a point $(x, x')$ is transported by $T$ by a distance no further than $\epsilon$. Let $\mu_0$ ad $\nu_0$ be the probability measures corresponding to the first and second marginals of $T_{\sharp \gamma}$ respectively. Then, $W_\infty(\mu, \mu_0) \le \epsilon$ and $W_\infty(\nu, \nu_0) \le \epsilon$. Hence,

$$\begin{aligned} D_\epsilon(\mu, \nu) &= \int_{\mathcal{X}^2} \mathbb{1}\{d(x, x') > 2\epsilon\} d\gamma \\ &= \int_{\mathcal{X}^2} \mathbb{1}\{d(x, x') > 0\} d\gamma_{\sharp T} \\ &\ge D_{TV}(\mu_0, \nu_0) \\ &\ge \inf_{W_\infty(p_0, p_0'), W_\infty(p_1, p_1') \le \epsilon} D_{TV}(\mu', \nu'). \end{aligned}$$

Combining with the reverse inequality that we proved above, it is clear that the infimum over $D_{TV}$ is attained by $\mu_0$ and $\nu_0$. $\qquad \square$

*Proof of Lemma 7.1.* The first equality in (14) follows from Theorem 7.2. For the second equality, we have the following.

$$\inf_{\substack{q \in \overline{\mathcal{P}}(\mathcal{X}): \\ q \preceq Tp_0}} D_\epsilon(q, p_1) \overset{(i)}{=} 1 - (T+1) \inf_{A \in \mathcal{B}(\mathcal{X})} R_{\oplus \epsilon}(\ell_{0/1}, A)$$

$$\overset{(ii)}{=} 1 - (T+1) \inf_{A \in \mathcal{B}(\mathcal{X})} R_{\Gamma_\epsilon}(\ell_{0/1}, A)$$

$$= 1 - (T+1) \inf_{A \in \mathcal{B}(\mathcal{X})} \sup_{\substack{W_\infty(p_0, p_0') \leq \epsilon \\ W_\infty(p_1, p_1') \leq \epsilon}} r(A, p_0', p_1')$$

$$\overset{(iii)}{\leq} 1 - (T+1) \sup_{\substack{W_\infty(p_0, p_0') \leq \epsilon \\ W_\infty(p_1, p_1') \leq \epsilon}} \inf_{A \in \mathcal{B}(\mathcal{X})} r(A, p_0', p_1')$$

$$= \inf_{\substack{W_\infty(p_0, p_0') \leq \epsilon \\ W_\infty(p_1, p_1') \leq \epsilon}} [1 - (T+1) \inf_{A \in \mathcal{B}(\mathcal{X})} r(A, p_0', p_1')]$$

$$\overset{(iv)}{\leq} \inf_{\substack{W_\infty(p_0, p_0') \leq \epsilon \\ W_\infty(p_1, p_1') \leq \epsilon}} \inf_{\substack{q' \in \overline{\mathcal{P}}(\mathcal{X}): \\ q' \preceq Tp_0'}} D_{TV}(q', p_1'),$$

where $(i)$ follows from Theorem 6.2, $(ii)$ from Theorem 5.1, $(iii)$ from Lemma E.1, and $(iv)$ again from Theorem 6.2 with $\epsilon = 0$.

We will now show the inequality in the opposite direction. That is, we will show the following.

$$\inf_{\substack{q \in \overline{\mathcal{P}}(\mathcal{X}): \ W_\infty(q, q') \leq \epsilon \\ q \preceq Tp_0 \ W_\infty(p_1, p_1') \leq \epsilon}} D_{TV}(q', p_1') \geq \inf_{\substack{W_\infty(p_0, p_0') \leq \epsilon \\ W_\infty(p_1, p_1') \leq \epsilon}} \inf_{\substack{q' \in \overline{\mathcal{P}}(\mathcal{X}): \\ q' \preceq Tp_0'}} D_{TV}(q', p_1') \qquad (32)$$

Consider arbitrary probability measures $q', p_1' \in \overline{\mathcal{P}}(\mathcal{X})$ generated in accordance with the constraints over the infimum terms on the left hand side of the above inequality. That is, let $q'$ and $p_1'$ be such that $W_\infty(q, q') \leq \epsilon$ and $W_\infty(p_1, p_1') \leq \epsilon$ where $q \preceq Tp_0$. We will now construct $p_0' \in \overline{\mathcal{P}}(\mathcal{X})$ such that $q' \preceq Tp_0'$ and $W_\infty(p_0, p_0') \leq \epsilon$. This will show that the set of $q', p_1' \in \overline{\mathcal{P}}(\mathcal{X})$ satisfying the constraints over the infimum terms on the right hand side is a superset of the corresponding set on the right hand side, and hence prove the above inequality.

Define a probability measure $p_0' \in \overline{\mathcal{P}}(\mathcal{X})$ as $p_0'(A) = p_0(A) + \frac{1}{T}q'(A) - \frac{1}{T}q(A)$ for $A \in \mathcal{B}(\mathcal{X})$. To show that $p_0'$ is a valid probability measure, we have the following.

$$p_0'(\mathcal{X}) = p_0(\mathcal{X}) + \frac{1}{T}q'(\mathcal{X}) - \frac{1}{T}q(\mathcal{X}) = 1$$

$$p_0'(A) = \frac{1}{T}(Tp_0(A) - q(A)) + \frac{1}{T}q'(A) \geq \frac{1}{T}q'(A) \geq 0.$$

The above equality also shows that $q' \preceq Tp_0'$. We will now show that $W_\infty(p_0, p_0') \leq \epsilon$. Since $W_\infty(q, q') \leq \epsilon$, there exists $\gamma \in \Pi(q, q')$ such that $\gamma(\{(x, x') \in \mathcal{X}^2 : d(x, x') \leq 2\epsilon\}) = 1$. Define $\gamma' \in \Pi(p_0, p_0')$ as follows for $A \in \mathcal{B}(\mathcal{X}^2)$.

$$\gamma'(A) = p_0(\{x \in \mathcal{X} : (x, x) \in A\}) + \frac{1}{T}\gamma(A) - \frac{1}{T}q(\{x \in \mathcal{X} : (x, x) \in A\}).$$

To see that $\gamma' \in \Pi(p_0, p_0')$, we have the following for $A_1, A_2 \in \mathcal{B}(\mathcal{X})$.

$$\gamma'(A_1 \times \mathcal{X}) = p_0(A_1) + \frac{1}{T}q(A_1) - \frac{1}{T}q(A_1) = p_0(A_1),$$

$$\gamma'(\mathcal{X} \times A_2) = p_0(A_2) + \frac{1}{T}q'(A_2) - \frac{1}{T}q(A_2) = p_0'(A_2).$$

Moreover,

$$\gamma'(\{(x, x') \in \mathcal{X}^2 : d(x, x') \leq 2\epsilon\}) = p_0(\mathcal{X}) + \frac{1}{T}\gamma(\{(x, x') \in \mathcal{X}^2 : d(x, x') \leq 2\epsilon\}) - \frac{1}{T}q(\mathcal{X}) = 1.$$

Therefore, $W_\infty(p_0, p_0') \leq \epsilon$. $\qquad \square$

*Proof of Theorem 7.4.* Without loss of generality, we assume $T \geq 1$. (If $T < 1$, we simply repeat the proof with labels $0$ and $1$ swapped.)

$$\inf_{\substack{A \in \mathcal{B}(\mathcal{X})}} \sup_{\substack{W_\infty(p_0, p_0') \leq \epsilon \\ W_\infty(p_1, p_1') \leq \epsilon}} r(A, p_0', p_1') = \inf_{A \in \mathcal{B}(\mathcal{X})} R_{\Gamma_\epsilon}(\ell_{0/1}, A)$$

$$\overset{(i)}{=} \inf_{A \in \mathcal{B}(\mathcal{X})} R_{\oplus \epsilon}(\ell_{0/1}, A)$$

$$\overset{(ii)}{=} \frac{1}{T+1} \left[ 1 - \inf_{\substack{q \in \overline{\mathcal{P}}(\mathcal{X}): \\ q \preceq T p_0}} D_\epsilon(p_0, p_1) \right]$$

$$\overset{(iii)}{=} \frac{1}{T+1} \left[ 1 - \inf_{\substack{W_\infty(p_0, p_0') \leq \epsilon \\ W_\infty(p_1, p_1') \leq \epsilon}} \inf_{\substack{q' \in \overline{\mathcal{P}}(\mathcal{X}): \\ q' \preceq T p_0'}} D_{TV}(q', p_1') \right]$$

$$= \sup_{\substack{W_\infty(p_0, p_0') \leq \epsilon \\ W_\infty(p_1, p_1') \leq \epsilon}} \frac{1}{T+1} \left[ 1 - \inf_{\substack{q' \in \overline{\mathcal{P}}(\mathcal{X}): \\ q' \preceq T p_0'}} D_{TV}(q', p_1') \right]$$

$$\overset{(iv)}{=} \sup_{\substack{W_\infty(p_0, p_0') \leq \epsilon \\ W_\infty(p_1, p_1') \leq \epsilon}} \inf_{A \in \mathcal{B}(\mathcal{X})} r(A, p_0', p_1'),$$

where (i) follows from Theorem 5.1, (ii) from Theorem 6.2, (iii) from Lemma 7.1 and (iv) follows again from Theorem 6.2 with $\epsilon = 0$. $\qquad \square$