# OpenReview forum: "The Many Faces of Adversarial Risk"
_NeurIPS.cc/2021/Conference — NeurIPS 2021 Poster_

### Official Review · Reviewer_pEiS · 2021-06-29

**Rating:** 6
**Confidence:** 4

**Summary:**

The paper discusses different definitions of adversarial risk that have been used in various papers throughout the years, where, by using tools from optimal transport theory, robust statistics, functional analysis, and game theory, these definitions are viewed under a different lens, and in several ways new connections (and separations) between some of these definitions are established.


**Limitations And Societal Impact:**

I am happy with the discussion that the authors have done in the paper.

**Main Review:**

The paper first gathers and discusses different notions of adversarial risk and then takes a unified approach on looking at these definitions in different settings. Though I have to admit that I have not carefully looked into the proofs, I think that for the largest part the paper is ok and provides the reader with new insights connecting different definitions of adversarial risk. Having said that, there are some claims in the related literature that are a little bit confusing to the reader and indeed not all the definitions are discussed.

Section 3 is very important as the authors discuss 4 different definitions of adversarial risk. However, the way some definitions are presented is confusing for the reader.
- For example, the authors claim that the formulation of adversarial risk studied in [9, 16] is a special case of the approach followed in [25, 34]. This is only partially true as [9] and [16] spend a significant amount explaining the differences between different definitions and in particular those that are used in [25] and [34]. So, while the papers [9] and [16] indeed discuss, among other things, a special case of the definitions used in [25] and [34], they nevertheless take a different route and explain that when misclassification is the goal for the generation of adversarial examples, then the definitions used in [25] and [34] may in turn compute completely wrong values. As an extreme case, for example, when PAC learning Boolean functions, it is often-times the case that we learn the target function completely. Then, in such situations [9] and [16] claim that indeed the learnt model is infinitely robust since it cannot be fooled by any adversarial example. However, both [25] and [34] can somehow claim that they can find adversarial examples in these situations, by making the model flip the prediction label. However, when the ground truth also changes, then there is no misclassification -- which is really the whole point behind adversarial examples. This is also true in equations (2) and (3) in this paper where we can see that the adversarial risk would be positive, even if the model was entirely correct against a ground function that does the labeling.
- And to go one step further, equation (3) in the current paper still makes this assumption, and I do not believe it is even studied in [26] which is cited as the paper where this definition has its origin. [26] uses the same definition of "error-region adversarial risk" that is used in [9] (or what is called "exact in the ball adversarial risk" in [16]).

The above points need to be clarified so that everything is crystal clear to the reader. The current paper studies variations of adversarial risk that follow along the lines of "corrupted-inputs adversarial risk" [9] or "constant in the ball adversarial risk" [16]. Such versions of risk are indeed special cases of [25] and [34]. But in general, and that is the main point of [9] and [16], these definitions do not compute, adversarial examples -- again, assuming that misclassification is the goal of an adversarial example, and I believe we all agree that this is indeed the goal.

Hence, as far as I understand, the current paper indeed tries to formulate variations of the "corrupted-inputs / constant in the ball" adversarial risk. This has indeed its own merit, but the presentation in section 3 needs to be more careful and really explain things in a better way. The current paper does not touch upon the different definitions that are out there -- in fact, in some situations I believe that the presentation is entirely misleading.

Getting along I must admit that I really appreciate all the effort and the work that the authors have done and indeed I find the different connections that are established, to be interesting. The authors try to be clear and make interesting comments on their results. Nevertheless, it is also a little bit unclear to me up to what point we can accept some of the consequences, in general, as this assumption of maintaining the ground-truth label in some region around a sampled point does not necessarily hold.

*Post-rebuttal:*

Thank you for the response to all the reviews. I am looking forward to seeing the improvements that have been suggested on clarifications by myself as well as from the other reviewers. Assuming these clarifications will be seen in the final manuscript, I am increasing my score from 5 to 6. Thank you for a very interesting paper!

**Time Spent Reviewing:**

5

---

> ### Author Response · Authors · 2021-08-09
> **On clarifying the discussion in Section 3 and other notions of adversarial risk**
>
> Thank you for your thoughtful review. We are glad that you "find the different connections that are established, to be interesting" and think that our paper "provides the reader with new insights connecting different definitions of adversarial risk". Regarding your comments on our presentation of adversarial risk definitions in section 3, we sincerely believe that we can address all your comments satisfactorily. Below, we state our explanations.
>
> > "For example, the authors claim that the formulation of adversarial risk studied in [9, 16] is a special case of the approach followed in [25, 34]. This is only partially true as [9] and [16] spend a significant amount explaining the differences between different definitions and in particular those that are used in [25] and [34]. So, while the papers [9] and [16] indeed discuss, among other things, a special case of the definitions used in [25] and [34], they nevertheless take a different route and explain that when misclassification is the goal for the generation of adversarial examples, then the definitions used in [25] and [34] may in turn compute completely wrong values."
>
> We do not claim that the formulation of adversarial risk studied in [9, 16] is a special case of the approach followed in [25, 34]. Such a claim would only be partially true like you stated, because [9, 16] survey different types of adversarial risk and focus on the "exact-in-the-ball" risk [16] or "error-region" risk [9], which is different from our formulation in equation (2). What we claim in Lines 113-114 is the following the following two things combined in a compound sentence: 1) The formulation in equation (2) has been studied in [9, 16], and 2) the formulation in equation (2) is a special case of the widely used expected supremum formulation of adversarial risk for general loss functions used in [25, 34].
>
> We will change the wording in Lines 113-114 so that there is no room for confusion.
>
> > "As an extreme case, for example, when PAC learning Boolean functions, it is often-times the case that we learn the target function completely. Then, in such situations [9] and [16] claim that indeed the learnt model is infinitely robust since it cannot be fooled by any adversarial example. However, both [25] and [34] can somehow claim that they can find adversarial examples in these situations, by making the model flip the prediction label. However, when the ground truth also changes, then there is no misclassification -- which is really the whole point behind adversarial examples. This is also true in equations (2) and (3) in this paper where we can see that the adversarial risk would be positive, even if the model was entirely correct against a ground function that does the labeling."
>
> You are absolutely right in noticing that our formulations in equations (2) and (3) can give a positive adversarial risk even for a "perfect" classifier. This is consistent with the literature on adversarial examples where even a perfect classifier (for example, the Bayes classifier in binary classification with 0-1 loss setting) is forced to make errors in the presence of evasion attacks. Our formulations correspond to "constant-in-the-ball" risk in [16] or "corrupted-instance" risk in [9, 26]. Our formulations are motivated from the study on mitigating adversarial attacks and the resulting notions of risk in [25, 34].
>
> In our formulation, an adversarial risk of zero is only possible if the supports of the true data distributions are non-overlapping and separated by at least $2\epsilon$. If the supports are not disjoint, then perfect robust learning is impossible. Such a formulation is relevant in understanding fundamental accuracy limits imposed by real-world adversarial examples. For instance, it is shown in Elsayed et al. (NeurIPS 2018) that it is possible to construct adversarial attacks that fool both computer vision and time-limited humans. In a use-case like collision detection for autonomous vehicles, it is not unreasonable to think of the true distributions as having overlapping support due to the confusion factor in time-limited humans, making perfect learning impossible.
>
> We acknowledge the value in studying "exact-in-the-ball" risk [16] or "error-region" risk [9]. However, these formulations of adversarial risk are not the focus of our paper. All the definitions presented in section 3 correspond to "corrupted-instance" family of adversarial risks.
>
> We will add to the discussion in section 3 to make this distinction clear.
>
>
> > "And to go one step further, equation (3) in the current paper still makes this assumption, and I do not believe it is even studied in [26] which is cited as the paper where this definition has its origin. [26] uses the same definition of "error-region adversarial risk" that is used in [9] (or what is called "exact in the ball adversarial risk" in [16])."
>
> Our claim is *not* that the formulation in equation (3) originated from [26]. In Line 119, we claim that an alternate formulation of adversarial risk (shown in equation (3)) replaces the Minkowski set expansion (which is used in equation (2)) by closed set expansion. The b-expansion of a set in Definition 2.2 in [26] is identical to our $\epsilon$-closed expansion in Lines 87-88. This notion of set expansion does not have measurability problems in any "nice" metric probability space, like a Polish space. It is also stated in [26] that this condition is needed to define adversarial risk in Definition 3.1. However, as we discuss in Lines 121-123, the $\epsilon$-closed expansion formulation may violate the $\epsilon$ distance budget constraint for the adversary at some points $x$.
>
> Although the main focus of [26] is the study of "error-region" risk, they do discuss other notions like "prediction change" risk and "corrupted-instance" risk. The point we tried to make in Line 119 was that [26] uses the closed set expansion rather than the (more appropriate) Minkowski set expansion in order to overcome measurability issues. Equation (3) is what we get if we apply the same simplification to equation (2).
>
> We will change the wording in Line 119 so that there is no room for confusion.
>
> > "Hence, as far as I understand, the current paper indeed tries to formulate variations of the "corrupted-inputs / constant in the ball" adversarial risk. This has indeed its own merit, but the presentation in section 3 needs to be more careful and really explain things in a better way. The current paper does not touch upon the different definitions that are out there -- in fact, in some situations I believe that the presentation is entirely misleading."
>
> You are right in observing that our study focuses on "corrupted-inputs / constant in the ball" adversarial risk. We have not discussed "exact in the ball" notions of risk in more detail in an effort to not over-burden the reader with more definitions early on. We are happy to include a discussion on these other notions of adversarial risk under related work. We hope our comments above help make our presentation more clear.
>
> > "Getting along I must admit that I really appreciate all the effort and the work that the authors have done and indeed I find the different connections that are established, to be interesting. The authors try to be clear and make interesting comments on their results. Nevertheless, it is also a little bit unclear to me up to what point we can accept some of the consequences, in general, as this assumption of maintaining the ground-truth label in some region around a sampled point does not necessarily hold."
>
> Thank you for your kind comments. Regarding the implicit assumption with "prediction-change" risk: As remarked in [9], the assumption of maintaining the ground-truth label in some region around a sampled point does hold in natural settings like image classification. Adding an imperceptible perturbation to an image of a cat does not change the ground-truth label because a human would still be able to classify the image as a cat. Also note that our formulation of adversarial risk is "corrupted-inputs" risk like the one studied in [25, 34] and not the "prediction-change" risk. In our formulation, there is no assumption that the ground-truth label is "smooth" (i.e. constant in a ball). In fact, the ground truth label is random even at a fixed $x$.

---

### Official Review · Reviewer_A5zG · 2021-07-11

**Rating:** 6
**Confidence:** 3

**Summary:**

This paper provides a lot of theoretical insights to connect the different definitions of adversarial risk. It also discusses some other properties such as well-definedness and minimax theorem.

**Limitations And Societal Impact:**

No social impact.

**Main Review:**

Originality: this paper utilizes a lot of different mathematical tools and provides rigorous analysis. Such an analysis is complicated and is novel based on my knowledge.

Quality, clarity and significance: my biggest concern is that the paper piles up a lot of different definitions and mathematical symbols, which may prevent readers from related areas to understand it (e.g. people who work on empirical study of adversarial training).  The authors are strongly encouraged to reorganize this paper and simplify some notations so that it has a more clear structure and is easier to understand.  Two examples:

(1) Although this paper uses the definition of adversarial risk in [9], the definition in [9] is self is much easier than the one in this paper to understand. Please consider use simple definitions first (even informal), and try to provide their more formal definitions later.

(2) The four contributions listed in Section 1 need to have a clear connection with the following five sections in this paper. I would suggest to first write down the corresponding sections when listing the contributions, and recall the readers about the this when starting each section.

The authors also need to emphasize the significance of this paper, e.g. provide some concrete easy-to-understand example where some definitions of adversarial risk may fail while others work well, and provide some experiments (or even simple simulation) to help illustration. It is hard to justify the significance of this paper from its current writing.

A minor issue:
A missing related reference: Xu, Q., Bello, K., & Honorio, J. (2020). A Le Cam Type Bound for Adversarial Learning and Applications. arXiv preprint arXiv:2007.00289.

====================================================

After author feedback:

The authors provided useful materials for the readability based on my major concern. Raised my score from 5 to 6.




**Time Spent Reviewing:**

5

---

> ### Author Response · Authors · 2021-08-09
> **On clarity and significance of our results**
>
> Thank you for your review. We are glad that you think our paper "is novel", "provides a lot of theoretical insights to connect the different definitions of adversarial risk" and "provides rigorous analysis". Below, we answer your concerns.
>
> ## On Quality and Clarity:
>
> > "My biggest concern is that the paper piles up a lot of different definitions and mathematical symbols, which may prevent readers from related areas to understand it"
>
> Exploring "the many faces" of adversarial risk is one of the main objectives of the paper, and hence the plethora of definitions up front seems unavoidable. However, we do agree with you that it makes it hard for readers from empirical backrgounds. We will revise our exposition to make it easier to read for broader audience by including more examples, figures and presenting simpler definitions first wherever possible.
>
> > "(1) Although this paper uses the definition of adversarial risk in [9], the definition in [9] is self is much easier than the one in this paper to understand. Please consider use simple definitions first (even informal), and try to provide their more formal definitions later."
>
> We will take the following action to make our submission more readable to empirical researchers: Since adversarial risk is commonly defined for general loss function in empirical works (like in [9]), we will start section 3 with the general case, and then specialize to 0-1 loss.
>
> > "(2) The four contributions listed in Section 1 need to have a clear connection with the following five sections in this paper. I would suggest to first write down the corresponding sections when listing the contributions, and recall the readers about the this when starting each section."
>
> We respectfully disagree; the four contributions in Section 1 clearly identify their location in the paper.
>
> ## On Significance:
>
> > "The authors also need to emphasize the significance of this paper, e.g. provide some concrete easy-to-understand example where some definitions of adversarial risk may fail while others work well, and provide some experiments (or even simple simulation) to help illustration."
>
> We will expand on the measurability example given in Line 99 by adding it to the appendix. We will include figures to better illustrate various definitions presented in section 3.
>
> > "It is hard to justify the significance of this paper from its current writing."
>
> We respectfully disagree. Indeed, our paper already shows the benefits of building a solid foundation for empirical researchers:
> 1. Algorithms for set-based risk can be very different from those for $W_\infty$-ball risk, but our equivalency results show that the desired robustness is achieved by both routes.
> 2. Our minimax results show adversarial classifiers are simply Bayes classifiers for the worst case distributions. Could one design algorithms that identify the worst-case distributions? This would be an entirely new approach to getting robust algorithms.
> 3. Our connection to the theory of capacities could allow for extensions to diverse classes of adversaries, such as total variation/Huber's/Prokhorov distance contaminations.
>
> Thank you for the reference of Xu et al. (2020). We will include it in our additional discussion of related work.

---

> > ### Comment · Reviewer_A5zG · 2021-08-12
> > **Comment**
> >
> > Thanks for the explanation. You mentioned that "We will take the following steps to make our paper more readable to a broader audience. Since the adversarial risk definition with general loss function is more familiar to applied researchers, we will introduce the discussion in section 3 with such a definition and then specialize to 0-1 loss. We will add figures to better illustrate various definitions presented in section 3." Could you have some details here?

---

> > > ### Author Response · Authors · 2021-08-14
> > > **Details on how we will make our paper more readable to a broader audience**
> > >
> > > Thank you for your comment. We really appreciate your willingness to engage with us in an effort to improve our paper. Below, we provide details on how we will make our paper more readable to a broader audience.
> > >
> > > ### Introducing the definition of adversarial risk
> > >
> > > We will start section 3 with the following introduction.
> > >
> > > > For a space of classifiers parametrized by $w\in \mathcal{W}$ and a loss function $\ell: (\mathcal{X}\times\mathcal{Y})\times \mathcal{W}\to [0, \infty)$, the adversarial risk of a classifier $w\in \mathcal{W}$ under an adversarial budget of $\epsilon\geq 0$ is defined as,
> > > \begin{align}
> > >     R_\epsilon(\ell, w) = E_{(x,y)}\left[ \sup_{d(x,x')\leq \epsilon} \ell((x', y),w) \right].
> > > \end{align}
> > > A case of special interest is the $0$-$1$ loss function with *non-parametric* classifiers of the form $f_A(x) = 1\\{x\in A\\}$ where $A\in \sigma(\mathcal{X})$, the set of all measurable subsets of $\mathcal{X}$. In this case, $\ell_{0/1}((x,y), A) = 1\\{x\in A, y = 0\\} + 1\\{x\in A^c, y = 1\\}$, and we have the following.
> > > \begin{align*}
> > >     R_\epsilon(\ell_{0/1}, A)
> > >     &= \frac{T}{T+1} E_{p_0}\left[ \sup_{d(x,x')\leq \epsilon} 1\\{x'\in A\\} \right]
> > >     + \frac{1}{T+1} E_{p_1}\left[ \sup_{d(x,x')\leq \epsilon} 1\\{x'\in A^c\\} \right]\\\\
> > >     &= \frac{T}{T+1} p_0(A^{\oplus\epsilon}) + \frac{1}{T+1} p_1((A^c)^{\oplus\epsilon}).
> > > \end{align*}
> > >
> > > Since adversarial risk is more commonly presented in the $E\[\sup \ell(\cdot)\]$ formulation [25, 34], we hope this introduction helps make our paper more readable to applied researchers.
> > >
> > > ### Adding figures to illustrate definitions
> > >
> > > We will add a figure to section 3 to explain the differences between the four definitions of adversarial risk ($R_{\oplus\epsilon}, R_{\epsilon}, R_{F_\epsilon}, R_{\Gamma_\epsilon}$) introduced in section 3. The figure will have 4 sub-figures, each focusing on a particular definition of adversarial risk. We are unable to include the figure with this comment here, but it will illustrate the following.
> > >
> > > - For $R_{\oplus\epsilon}$, the adversary can perturb any particular $x$ to $B_\epsilon(x)$.
> > > - For $R_{\epsilon}$, the adversary can perturb an $x$ in any set $A$ into the set $A^\epsilon$.
> > > - For $R_{F_\epsilon}$, the adversary can perturb any $x$ to $f(x)$ with any measurable function $f: \mathcal{X}\to \mathcal{X}$.
> > > - For $R_{\Gamma_\epsilon}$, the adversary can perturb the true distributions $p_0, p_1$ to any $p_0', p_1'$ such that $W_\infty(p_i, p_1')\leq \epsilon$ for $i=0,1$.
> > >
> > > We also have figure ideas for illustrating some of our key results, but we are limited by the space constraint. We welcome any other suggestions from you that will improve the readability of our paper.
> > >
> > > We hope we answered your query to your satisfaction. And of course, we are happy to continue this discussion further.

---

### Official Review · Reviewer_YZqf · 2021-07-12

**Rating:** 6
**Confidence:** 4

**Summary:**

The paper extends previous works of the study of the adversarial risk. The authors interest in the well definition of the adversarial risk. They then show a minimax theorem on the game between the classifier and the adversary.

**Limitations And Societal Impact:**

The understanding of Adversarial robustness is a very important topic. It can impact positively the safety of autonomous ML systems.

**Main Review:**

Overall I enjoyed reading the paper. The paper brings insights on Adversarial risks extending previous results from the literature. The paper is well written and easy to follow. The current trend of theoretical papers on adversarial attacks can be only beneficial for the understanding of the topic. I tried to make a thorough reading of the proofs, but I still might have missed some points. I would like to discuss the following points with the authors:

- On the definition of the adversarial risk. I believe it is provable that $A^{\oplus\epsilon}$ is  analytic (cf 1 for more details) and hence universally measurable. Idea: write $1_{x\in A^{\oplus\epsilon}}$ as $\sup_{a\in A} 1_{x\in\bar{B}_\epsilon(a)}$. Then you can use the results from Bertsekas and Shreve (2004), Prop 7.39 and Corollary 7.42.1. Maybe the authors can use these ideas to prove a more general result. I think all further results can be extended to all Borel sets instead that only $G_\delta$ and $F_\sigma$ sets. I am happy to discuss more about this during the rebuttal phase. I might be mistaken, but I think it worth discussing. It would simplify some results in the paper I’d say.

- The authors said they prove the existence of Pure Nash equilibria. Although the classifier is deterministic, the attacker is still « randomized ». Maybe they should mention that the equilibria are hybrid. This result seems right and reasonable.

- I would also suggest the authors to give some more intuition to the different notions of risks the authors introduced. For instance, draw some figures to explain it. It would be more readable for non advertised readers

- Finally, although I enjoyed reading the paper, I also find that the paper lacks a bit of novelty with regards to existing literature (Pydi et al., Dohmatob, Bhagoji et al.). These papers have proved the majority of the results exposed in the paper ($W_\infty$, $c_\epsilon$ costs,etc.). The authors managed to treat unbalanced probabilities. This is my major concern about accepting/rejecting the paper. I am willing to discuss with the authors and the other reviewers on this.

For now, I rate this paper as borderline,  but I am willing to change my rating if the authors explain to me more clearly the novelties of their paper. Moreover, I hope authors can get rid of their hypothesis on $F_\sigma$ and $G_\delta$.

**Time Spent Reviewing:**

3

---

> ### Author Response · Authors · 2021-08-09
> **On possible simplifications, novelty and other comments**
>
> Thank you for your thoughtful review. We are glad that you enjoyed reading our paper and stated that our paper "brings insights on Adversarial risks extending previous results from the literature".  Regarding your comment, "For now, I rate this paper as borderline, but I am willing to change my rating if the authors explain to me more clearly the novelties of their paper", we sincerely believe that we can address all your comments satisfactorily. Below, we address your concerns.
>
> ## On Possible Simplifications
>
> > "On the definition of the adversarial risk. I believe it is provable that $A^{\oplus\epsilon}$ is analytic (cf 1 for more details) and hence universally measurable. Idea: write $1_{x\in A^{\oplus\epsilon}}$ as $\sup_{a\in A} 1_{x\in B_\epsilon(a)}$. Then you can use the results from Bertsekas and Shreve (2004), Prop 7.39 and Corollary 7.42.1. Maybe the authors can use these ideas to prove a more general result."
>
> Yes, **we already explored this line of attack but it does not help in simplifying our results**. The strategy that you mention was used in Lemma 4 in Meunier et al. (ICML 2021)[32] who also use Prop 7.39 and Corollary 7.42.1. from Bertsekas and Shreve (2004) to prove the well-definedness of adversarial risk for upper semi-continuous loss functions. The equivalent to an upper semi-continuous loss function in our setting is a decision region $A$ that is a closed set under 0-1 loss function. (Here, note that the indicator function of a closed set is upper semi-continuous.) Hence, we can use this strategy to prove that $A^{\oplus\epsilon}$ is analytic (and hence universally measurable) whenever $A$ is closed. To see this, consider that $A^{\oplus\epsilon} = Proj_2\\{ (x,x')\in \mathcal{X}^2 | 1\\{(x,x'): x\in A\\} - c_\epsilon(x,x')>0 \\}$ where $c_\epsilon$ is the cost as defined in [32]. If $A$ is a closed set, then $1\\{(x,x'): x\in A\\}$ is upper semi-continuous and it follows from Prop 7.39 and Corollary 7.42.1. from Bertsekas and Shreve (2004) that $A^{\oplus\epsilon}$ is Borel measurable whenever $A$ is closed. But a stronger result already appears in Pydi and Jog (ICML 2020)[32] who show that $A^{\oplus\epsilon}$ is closed (not just Borel measurable) whenever $A$ is closed. In this paper, we further relax the closed set condition to the much broader class of $G_\delta$ and $F_\sigma$ sets, which includes all closed sets, all open sets, and any countable union of closed or open sets.
>
> > "I think all further results can be extended to all Borel sets instead that only $F_\sigma$ and $G_\delta$ sets. I am happy to discuss more about this during the rebuttal phase. I might be mistaken, but I think it worth discussing. It would simplify some results in the paper I’d say."
>
> For $\mathbb{R}^d$, our results are applicable to all Lebesgue measurable sets (which includes all Borel sets). For general Polish spaces, **there are strong reasons to believe that our results cannot be extended further to all Borel sets** instead that only $F_\sigma$ and $G_\delta$ sets. We state them below.
> 1. Theorem 5.1 relies crucially on the Kuratowski-Ryll-Nardzewski (KRN) measurable selection theorem to build a measurable map for the adversary. For KRN theorem (or in general, any type of measurable selection theorem) to be applicable like in the proof of Theorem 5.1, we need $A$ to be Polish. Since $G_\delta$ subsets are equivalent to Polish subsets in a metric space (see Lemma 3.33 and Lemma 3.34 in [1]), we cannot relax the $G_\delta$ assumption.
> 2. We already have a counter example for a Borel set $A$ for which $A^{\oplus\epsilon}$ is not Borel. See Lines 92-93.
> 3. So far, the Borel measurability for $A^{\oplus\epsilon}$ has only been proven for $F_\sigma$ sets. See Lines 155-157.
>
> We briefly explain the need for $F_\sigma$ and $G_\delta$ assumption in Lines 202-204. We are happy to extend this discussion.
>
> ## On Novelty
>
> > "Finally, although I enjoyed reading the paper, I also find that the paper lacks a bit of novelty with regards to existing literature (Pydi et al., Dohmatob, Bhagoji et al.). These papers have proved the majority of the results exposed in the paper ($W_\infty$, $c_\epsilon$ costs,etc.). The authors managed to treat unbalanced probabilities. This is my major concern about accepting/rejecting the paper. I am willing to discuss with the authors and the other reviewers on this."
>
> **The main novel contribution of our paper is the existence of Nash equilibrium in adversarial examples game for *non-parametric* classifiers.**
> - *Existing results* [27,31,3]: Meunier et al. (ICML 2021)[27] show "mixed" Nash equilibrium for *randomized* classifiers parametrized by points in a Polish space. Pinot et al. (ICML 2020)[31] and Bose et al. (NeurIPS 2020)[3] and  tackle the same problem but for a *regularized* adversary. The equilibrium analysis in these papers uses Fan's minimax theorem with concave-convex condition.
> - *Our result:* We consider *pure* classifiers that are *non-parametric*, expressed as decision regions. The non-parametric setting makes the analysis more challenging. **Fan's theorem is inapplicable in our setting** because concave/convex notions fall apart for sets. Instead, we use tools from Huber's 2-alternating capacities for $\mathbb{R}^d$. For Polish spaces, **we develop our tools "from scratch"** by generalizing Strassen's duality theorem and using optimal transport techniques. The connection with Huber's capacities (which we prove in Lemma 5.2) and the generalization of Strassen's theorem (Theorem 6.1) are both novel to the best of our knowledge.
>
> **Other novelties of our paper:**
> 1. Extending the results of Pydi and Jog (ICML 2020)[32] and Bhagoji et al. (NeurIPS 2019)[2] to unequal priors, by generalizing the classical Strassen's theorem to finite measures.
> 2. Equivalence of a particular notion of adversarial robustness with $\infty$-Wasserstein distributional robustness. This result does NOT appear in Pydi and Jog (ICML 2020)[32] or Bhagoji et al. (NeurIPS 2019)[2] or Dohmatob (ICML 2019, arxiv 2020)[10, 11]. A similar result appears in Tu et al. (NeurIPS 2019)[41] and Staib and Jegelka (NeurIPS Workshop 2017)[37], both of which gloss over measurability issues.
> 3. Relation between adversarial robustness and 2-alternating capacities. To the best of our knowledge, this connection has not been made before. This connection is crucial in proving the existence of a pure Nash equilibrium in $\mathbb{R}^d$. It may also be useful to extend our analysis to more general perturbation models.
> 4. Conditions under which various notions of adversarial risk are well-defined.
>
> ## On the "Purity" of the Nash Equilibrium
>
> > "The authors said they prove the existence of Pure Nash equilibria. Although the classifier is deterministic, the attacker is still « randomized ». Maybe they should mention that the equilibria are hybrid. This result seems right and reasonable."
>
> As stated at the start of section 7, the action space of the attacker is the $\epsilon$-ball of probability measures around the true distributions. **Within the action space, the optimal action is not randomized**. For instance, in Theorem 7.3, we show that there exists a single worst-case pair of probabilities $p_0^*, p_1^*$ and a single optimal classifier $A^*$ that constitutes the equilibrium point. Hence, we think "pure" Nash equilibrium is indeed the correct characterization as opposed to mixed/hybrid Nash equilibrium. **In contrast, the equilibrium established in Meunier et al. (ICML 2021)[27] consists of a *mixture* of classifiers (as opposed to a single classifier)** and a worst-case perturbed probability measure. We are happy to discuss this matter further.
>
> ## On Readability
>
> > "I would also suggest the authors to give some more intuition to the different notions of risks the authors introduced. For instance, draw some figures to explain it. It would be more readable for non advertised readers"
>
> Thank you for the suggestion. We will include figures to better illustrate various definitions presented in section 3.

---

> > ### Comment · Reviewer_YZqf · 2021-08-10
> > **Comments**
> >
> > Thanks for your answer! I really appreciate the detailed comments. There are still some points I might discuss with you.
> >
> > ## On extension to all Borel sets.
> >
> > In general, an upper analytic function is general. Maybe you can refer to the definitions and results from Bertsekas and Shreve. But it seems that Borel measurable functions are upper semi analytic. I think you might have missed that previous point. In Meunier et al., they used upper semi-continuity which implies Borel measurability. Hence I truly believe that $A^{\oplus\epsilon}$ is indeed analytic and then universally measurable. But, still, I might be wrong. If the authors think it is the case, please tell me!
> >
> > Concerning the extension of all the results, I can answer your intuitions:
> > 1. I still refer to Bertsekas and Shreve: Theorem 5.1 might be proven using prop 7.50. But once again, I am not 100% sure, but it seems reasonable that using this, Theorem 5.1 holds on a wider class of sets.
> > 2. I totally agree, the set $A^{\oplus\epsilon}$ should be analytic (and then universally measurable) which is more general than Borel sets
> > 3. The same argument that 2.
> >
> >
> > ## On novelty
> > Thanks for the clarification but I still have some concerns. I agree that most papers did not treat about measurability. $\infty$-Wasserstein distributional robustness actually appears in the updated arxiv version of Pydi and Jog. And the results about Nashequilibria seems to be a consequence of Dohmatob (arxiv 2020), even if it is not stated as it.
> >
> > ## On purity of Nash equilibria
> >
> > I totally agree that your classifier is "pure" (i.e. deterministic). I reiterate that your attacker can be indeed "random". For instance take $\epsilon=1$, $p_0=\delta_0$ and $p'_0 = \frac12(\delta_{-1/2}+\delta_{+1/2})$, then $W_\infty(p_0,p'_0)\leq \epsilon. Here such an attack moves randomly 0 to -1/2 or 1/2 w.p. 1/2. That's why I think you should speak of hybrid equilibrium.
> >
> > Be careful also, your infima and suprema might not be attained. Probably the sup is attained. But the infimum is much more difficult to prove it is the case.
> >
> > I hope the authors can answers my comments. I really enjoy reading the paper, I just think some points can be improved.

---

> > > ### Author Response · Authors · 2021-08-11
> > > **Clarifying our novelty, and possible extension to all Borel sets**
> > >
> > > > "I hope the authors can answers my comments. I really enjoy reading the paper, I just think some points can be improved."
> > >
> > > We are glad that you enjoyed reading our paper! We really appreciate your willingness to engage with us in an effort to improve our results. Below, we provide answers to all your comments.
> > >
> > > ## On Extension to all Borel Sets
> > >
> > > > "In general, an upper analytic function is general. Maybe you can refer to the definitions and results from Bertsekas and Shreve. But it seems that Borel measurable functions are upper semi analytic. I think you might have missed that previous point. In Meunier et al., they used upper semi-continuity which implies Borel measurability. Hence I truly believe that $A^{\oplus\epsilon}$ is indeed analytic and then universally measurable. But, still, I might be wrong. If the authors think it is the case, please tell me!"
> > >
> > > Yes, we did refer to the results in Chapter 7 of Bertsekas and Shereve in an effort to extend our measurability results further. But, they do not seem to help extend our results beyond what we already presented in our paper. As we explained in our previous comment, the strategy of Meunier et al. cannot be used for 0-1 loss function except when the set $A$ is closed. As you already state in your comment, Meunier et al's strategy relies on the upper semi-continuity (u.s.c.) of the loss function (coupled with the lower semi-continuity of the $c_\epsilon$ cost) to show Borel measurability. But the 0-1 loss function need not be u.s.c. unless the decision region $A$ is closed. And the Borel measurability of $A^{\oplus\epsilon}$ for closed sets has already been established in Pydi and Jog (2020).
> > >
> > > ### Regarding $A^{\oplus\epsilon}$ being analytic:
> > >
> > > Yes it may be the case that $A^{\oplus\epsilon}$ is analytic whenever $A$ is Borel, but we are unable to verify this for a general Polish space.
> > >
> > > For $\mathbb{R}^d$, here is a simple argument to show that $A^{\oplus\epsilon}$ is indeed analytic whenever $A$ is Borel: Define the continuous map $f: \mathcal{X}^2\to \mathcal{X}$ as $f(x,x') = x+x'$. Then $A^{\oplus\epsilon} = f(A \times B_\epsilon(0))$. If $A$ is Borel, then $A \times B_\epsilon(0)$ is also Borel and hence, $A^{\oplus\epsilon}$ is analytic because it is a continous image of the Borel set $A \times B_\epsilon(0)$. Theorem 4.2 in our paper proves an even stronger statement than what this simple argument implies: we show that $A^{\oplus\epsilon}$ is Lebesgue measurable (i.e. universally measurable) whenever $A$ is Lebesgue measurable (not just Borel measurable).
> > >
> > > The above simple strategy does not work in general Polish spaces because of the lack of vector space structure. In $\mathbb{R}^d$, $A^{\oplus\epsilon} = A + B_\epsilon(0)$, but in a general Polish space, $A^{\oplus\epsilon} = \cup_{a\in A} B_\epsilon(a)$, a possibly *uncountable* union of closed sets.
> > >
> > > ### Regarding Prop 7.50 in Bertsekas and Shreve:
> > >
> > > Unfortunately, Prop 7.50 in Bertsekas and Shreve also *does not help* in showing that $A^{\oplus\epsilon}$ is analytic. Define the set $D\subset \mathcal{X}^2$ as $D = \\{(x,y): x\in A, d(x,x')\leq \epsilon\\}$ and the function $f: D\to \mathbb{R}^*$ as $f(x,y) = -1\\{x\in A\\}$. To apply Prop 7.50, we need to show that $D$ is analytic, which is the challenging bit! It is not clear why $D$ should be analytic except for the specific case when $\mathcal{X} = \mathbb{R}^d$.
> > >
> > > ## On Novelty
> > >
> > > Yes, we are aware of the arxiv update to Pydi and Jog (which later appeared on the IEEE Transactions on Information Theory [here](https://ieeexplore.ieee.org/document/9496634)) and Dohmatob (arxiv Nov 2020). We sincerely believe that **our work has *considerable* novelty over both of these works**. Below, we explain our novelty over these works:
> > >
> > > ### Pydi and Jog (arxiv Dec 2020)
> > >
> > > **Background:** Pydi and Jog study "Markov kernel" adversarial risk for *parametrized* classifiers (denoted by $R_{K_\epsilon}(\ell, w)$), where the adversary uses a collection of randomized maps (kernels) to perturb data points. It is easy to see that their definition falls in between the deterministic "transport map" adversarial risk in equation (4) (also studied in Pinot et al. ) and the $W_\infty$ adversarial risk. That is, $R_{F_\epsilon}(\ell, w) \leq R_{K_\epsilon}(\ell, w)\leq R_{\Gamma_\epsilon}(\ell, w)$.
> > >
> > > **Their result:** Theorem 1 of Pydi and Jog proves that $R_{K_\epsilon}(\ell, w) = R_{\Gamma_\epsilon}(\ell, w)$. It does not deal with the $R_{\oplus\epsilon}$ definition, which is the equivalent to the standard $\mathbb{E}[\sup \ell(\cdot)]$ adversarial risk in the non-parametric setting.
> > >
> > > **Our result:** Our Theorem 5.1 proves $R_{\oplus\epsilon}(\ell_{0/1}, A) = R_{F_\epsilon}(\ell_{0/1}, A) = R_{\Gamma_\epsilon}(\ell_{0/1}, A)$.
> > >
> > > **The novelty:**
> > >
> > > - **Our theorem shows the equivalence between *all four* formulations of adversarial risk that appear in the literature so far** ($R_{\oplus\epsilon}, R_{F_\epsilon}, R_{K_\epsilon}, R_{\Gamma_\epsilon}$), once we combine with the trivial inequality $R_{F_\epsilon} \leq R_{K_\epsilon}\leq R_{\Gamma_\epsilon}$.
> > > - **Our result is in the more challenging *non-parametric* setting with 0-1 loss**, where the result of Pydi and Jog does not apply.
> > > - **Pydi and Jog do not analyze $R_{\oplus\epsilon}$**, which is the non-parametric analog for the $\mathbb{E}[\sup \ell(\cdot)]$ type formulation of adversarial risk, the most commonly studied form of adversarial risk [25, 34].
> > >
> > >
> > > ### Dohmatob (arxiv Nov 2020)
> > >
> > > **Background:** As you noted in your comment, the results of Dohmatob as stated, do not talk about Nash equilibria. They do have a result (Theorem 3.2 in their paper) which is similar to our Theorem 7.2.
> > >
> > > **The novelty:**
> > >
> > > - **Dohmatob's result applies only for equal priors** in Polish spaces with mid-point property. We prove analogous results for unequal priors using an unbalanced optimal transport formulation.
> > > - **Even for equal priors, the existence of Nash equilibrium is *not* a *direct* consequence of the result of Dohmatom**. In the proof of Theorem 7.3, we crucially needed Theorem 5.1, showing the equivalence between $R_{\oplus\epsilon}$ and $R_{\Gamma_\epsilon}$ in order to make the inf-sup switch. As explained above, Theorem 5.1 is novel, and does not appear even in the updated arxiv version of Pydi and Jog.
> > > - **Extending the Nash equilibrium result for unequal priors is *non-trivial*** and relies on our unbalanced optimal transport formulation for adversarial risk proven in section 6.
> > > - **Our Nash equilibrium result in $\mathbb{R}^d$ uses an *entirely different* proof strategy** that builds on connections with 2-alternating capacities, and also applies for a much broader class of sets compared to the general Polish space setting.
> > >
> > > ## On Purity of Nash Equilibria
> > >
> > > Yes, we take your point that our attacker can indeed be random in the sense that they can perturb a single point $x$ to a distribution of points. We will mention this in our discussion on the equilibrium in section 7.
> > >
> > > > "Be careful also, your infima and suprema might not be attained. Probably the sup is attained. But the infimum is much more difficult to prove it is the case."
> > >
> > > For $\mathbb{R}^d$, Theorem 7.1 shows that both the sup and inf are indeed attained. This result crucially relies on the connections we establish with 2-alternating capacities. For general Polish spaces, showing the attainment of sup is indeed much easier.
> > >
> > > Again, we thank you for providing thoughtful comments on our work and engaging in this dialogue. We hope we have provided satisfactory answers to all your comments. And of course, we are happy to continue this discussion further.

---

> > > > ### Comment · Reviewer_YZqf · 2021-08-12
> > > > **comment**
> > > >
> > > > In my opinion, $1_{x\in A\oplus\epsilon} = \sup_{a\in A} 1_{x\in\bar{B}_\epsilon(a)}$ = \sup_{a\in \mathcal{X}} 1_{x\in\bar{B}_\epsilon(a)}-\infty 1_{a\in A}$ so that Prop 7.50 in Bertsekas and Shreve helps concluding. But I might be wrong, once again. Such more general results would make the paper even more consistent.
> > > >
> > > > # On novelty
> > > >
> > > > After reading the literature again, I think there are some nice novelties in the paper that worth be published. So I am eager to raise my rating to 6.

---

> > > > > ### Author Response · Authors · 2021-08-15
> > > > > **Thank you, and clarification on Prop 7.50 in Bertsekas and Shreve**
> > > > >
> > > > > > "After reading the literature again, I think there are some nice novelties in the paper that worth be published. So I am eager to raise my rating to 6."
> > > > >
> > > > > Thank you for appreciating the novelties in our paper. We are pleased that you are willing to increase the rating of our paper.
> > > > >
> > > > > > "In my opinion, $1_{x\in A^{\oplus\epsilon}} = \sup_\{a\in A\} 1_{x\in\bar{B}\epsilon(a)} = \sup_{a\in \mathcal{X}} 1_{x\in\bar{B}\epsilon(a)}-\infty . 1\\{a\in A\\}$ so that Prop 7.50 in Bertsekas and Shreve helps concluding. But I might be wrong, once again."
> > > > >
> > > > > Perhaps you meant to say, $1_{x\in A\oplus\epsilon} = \sup_\{a\in A\} 1_{x\in\bar{B}\epsilon(a)} = \sup_{a\in \mathcal{X}} 1_{x\in\bar{B}\epsilon(a)}-\infty . 1\\{a\notin A\\}$. The hard part in applying Prop 7.50 is in verifying that the function  $f(a,x) = \sup_{a\in \mathcal{X}} 1_{x\in\bar{B}\epsilon(a)}-\infty . 1\\{a\notin A\\}$ is upper semianalytic. To verify that $f$ is upper semianalytic, we need to show that its level sets are analytic. **Showing that $f$ is upper semianalytic seems to be as hard as showing that $A^{\oplus\epsilon}$ itself is analytic**. Hence, we are not sure if Prop 7.50 can help.
> > > > >
> > > > > > "Such more general results would make the paper even more consistent."
> > > > >
> > > > > Yes, we absolutely agree that extending our results to all Borel sets would make our paper better. For this work, we do this for $\mathbb{R}^d$ using the porous set argument in section 4. For general Polish spaces, we extended it beyond the closed sets of Pydi and Jog (2020) to $F_\sigma$ and $G_\delta$ sets.
> > > > >
> > > > > Again, we thank you for being open to conversation, and for your thoughtful comments.

---

> > > > > > ### Comment · Reviewer_YZqf · 2021-08-17
> > > > > > **Additional comment**
> > > > > >
> > > > > > Actually the sum of two Borel measurable function is measurable. Moreover, a Borel measurable function is upper semi analytic so, it might help to conclude, no?

---

> > > > > > > ### Author Response · Authors · 2021-08-19
> > > > > > > **You are right, Prop 7.50 is applicable**
> > > > > > >
> > > > > > > Yes, you are right. It seems that Prop 7.50 can indeed be used to show that $A^{\oplus\epsilon}$ is analytic.   This considerably simplifies our assumptions on the set $A$ for the well-definedness of $R_{\oplus\epsilon}$. In fact, it seems that Prop 7.47 is sufficient to do so. Moreover, the universally measurable selection result of Prop 7.50 can perhaps be used to extend our results on equivalence with Wasserstein robustness to Borel sets as well. We will explore this further.
> > > > > > >
> > > > > > > We sincerely thank you for providing valuable feedback that helped us improve our submission.

---

### Official Review · Reviewer_nVTq · 2021-07-16

**Rating:** 6
**Confidence:** 4

**Summary:**

The paper focuses on formalizing the definition of "adversarial risk" which is risk under test-time perturbation attacks. The focus is on subtle measure theoretic issues that arise when the distributions are continuous.

**Limitations And Societal Impact:**

The paper is mainly theoretical and the long term impacts are positive.

**Main Review:**

Several definitions are given for adversarial risk in the literature. Most are not formal, and some are formal. The paper’s starting point is the formal ones. Yet, it revisits those definitions in the general case where distributions are not discrete, and hence subtle points arise about measurability of events and in particular whether or not certain probabilities associated with adversarial risk are always well defined. In particular, it is argued that some definitions use a union of Minkowski balls, which in general is not a measurable set. Other subtle points arise depending on what measure is used (e.g., Lebesgue or not). The paper also gives conditions under which: (1) various definitions become equivalent, (2) certain objects of interest exist and are characterized by intuitive formulations (e.g., optimal classifier under attack and Nash equilibrium in adversary’s game against algorithm designers).

As far as I understand, these points only arise when we pay attention to subtle aspects of formalism in a measure theoretic sense for general measures. So, this makes the work quite theoretical, as “in practice” (e.g., when we deal with *algorithmic* attacks and distributions), everything is discrete and none of the issues studied here arise. So, I don’t think the paper has a “practical” effect, as basically all the definitions that paper revisits are fine in the discrete setting and all the subtle points the paper mentions only show up when we go beyond the discrete case and start to care about measurability of sets in the continuous regime.


In fact, typically, by just assuming the space to be Polish these issues go away (e.g., see the last paragraph of page 11 here http://cgm.cs.mcgill.ca/~breed/conc/Talagrand.pdf) but evidently, in the context of adversarial risk and optimal adversarial strategies certain issues still exist, and the paper shows how to resolve them by carefully finding sufficient conditions.

The writing is very clear and precise and readable, and that is a big plus for a paper that has such a goal of formalizing notions.

Overall, I think the paper fills a gap in an important direction, even though this gap is purely theoretical and has no immediate impact on the “practical” side. Still it is important to have a general and precise theory for this phenomenon, and that could be important to practice also in the long run. Considering the quality and clarity of the writing and its content, I think the paper will be useful for the program and recommend acceptance.

Comments:

What is the main reason for focusing on binary classification? Any major limitations for larger label sets?

The term “decision region” is used in line 46 which is not clear. It is clarified later in line 109, but it is confusing the first time around.

The paper’s formulation of adversarial risk does not pay any attention to whether or not the perturbed point is in fact misclassified. Of course, if the perturbed point is “close” to the original point to the degree that the “truth” does not change, this would happen and this is how adversarial examples are perceived (“imperceptible” changes in a setting where humans are the judge for true labels). The issue is discussed here:
http://proceedings.mlr.press/v89/suggala19a/suggala19a.pdf
and the cited paper [9]. This aspect does not negatively affect this paper, becaue the paper's focus is mostly on "welldefinedness of events", yet it would be good to know how this can fit into this paper’s formal studies.

The letter “T” is used for both the probability weight and the transportation notation.

Line 156 : “It generalizes a similar result of [32] for closed and open sets. ” What is the key new insight? In general, it would be good to point out the new part of the generalized arguments.

Line 290: “Our minimax theorems establish such an equality.”
But in the cited paper [31] the authors say "Then we demonstrate, in Section 4, the non-existence of a Nash equilibrium in the deterministic setting of this game".
How is their result compatible with yours? Please clarify.

Line 362: “transport characterization adversarial risk” I think you need to add “of” after “characterization”


**Time Spent Reviewing:**

9

---

> ### Author Response · Authors · 2021-08-09
> **On measurability, practical significance and other comments**
>
> Thank you for your careful and detailed review of our work. We are glad that you found our writing to be "very clear and precise and readable, and that is a big plus for a paper that has such a goal of formalizing notions". We are encouraged by your assessment that "Overall, ... the paper fills a gap in an important direction" and your comment, "... it is important to have a general and precise theory for this phenomenon, and that could be important to practice also in the long run. Considering the quality and clarity of the writing and its content, I think the paper will be useful for the program and recommend acceptance." Below, we address your main concerns.
>
> ## On the Focus on Measure Theoretic Issues
>
> > "The focus is on subtle measure theoretic issues that arise when the distributions are continuous."
>
> We respectfully disagree. **Rectifying the measure theoretic issues with various definitions of adversarial risk is *one* of our many contributions, but *not* the main focus of our paper.** We start our presentation with this focus (in section 4) because otherwise we will not be able to rigorously present our results in the subsequent sections, namely: relation to robust hypothesis testing and Choquet capacities in section 5, generalizing the results of [32,2] in section 6, proving minimax theorems and existence of Nash equilibria and extending the results of [27,31,3] in section 7.
>
> > "In fact, typically, by just assuming the space to be Polish these issues go away (e.g., see the last paragraph of page 11 here http://cgm.cs.mcgill.ca/~breed/conc/Talagrand.pdf) but evidently, in the context of adversarial risk and optimal adversarial strategies certain issues still exist, and the paper shows how to resolve them by carefully finding sufficient conditions."
>
> As we state in Lines 92-94, the measure theoretic issues persist even in Polish spaces. As Talagrand's comment from your reference indicates, assuming that the decision regions are compact simplifies some considerations, but careful analysis is still needed to establish our results in sections 5, 6 and 7.
>
> ## On the "Practical" Effect of our Paper
>
> > "As far as I understand, these points only arise when we pay attention to subtle aspects of formalism in a measure theoretic sense for general measures. So, this makes the work quite theoretical, as “in practice” (e.g., when we deal with algorithmic attacks and distributions), everything is discrete and none of the issues studied here arise. So, I don’t think the paper has a “practical” effect, as basically all the definitions that paper revisits are fine in the discrete setting and all the subtle points the paper mentions only show up when we go beyond the discrete case and start to care about measurability of sets in the continuous regime."
>
> Measurability may not be a problem, but unclear definitions certainly can be. Indeed, our paper already shows the benefits of building a solid foundation for empirical researchers:
> 1. Algorithms for set-based risk can be very different from those for $W_\infty$-ball risk, but our equivalency results show that the desired robustness is achieved by both routes.
> 2. Our minimax results show adversarial classifiers are simply Bayes classifiers for the worst case distributions. Could one design algorithms that identify the worst-case distributions? This would be an entirely new approach to getting robust algorithms.
> 3. Our connection to the theory of capacities could allow for extensions to diverse classes of adversaries, such as total variation/Huber's/Prokhorov distance contaminations.
>
> ## Other Comments
>
> Here are our answers to your remaining questions:
>
> > "What is the main reason for focusing on binary classification? Any major limitations for larger label sets?"
>
> Understanding binary classification is the first step to understanding adversarial robustness in more complicated cases. A possible roadblock to multiclass classification could be extending Strassen's theorem in this setting. We have not explored how/if this can be extended to the case of more than 2 labels.
>
> > "Line 156 : “It generalizes a similar result of [32] for closed and open sets. ” What is the key new insight? In general, it would be good to point out the new part of the generalized arguments."
>
> The key hurdle was the measurability of $A^{\oplus \epsilon}$ -- the set expansion. When $A$ is open or closed, this fact is easy to prove (as in [32]). In this paper, we identify the broadest class of sets for which this holds. The preliminary lemmas in Appendix A of the supplementary pdf show how we use these properties to extend the analysis of [32].
>
> > "The paper’s formulation of adversarial risk does not pay any attention to whether or not the perturbed point is in fact misclassified. Of course, if the perturbed point is “close” to the original point to the degree that the “truth” does not change, this would happen and this is how adversarial examples are perceived (“imperceptible” changes in a setting where humans are the judge for true labels). The issue is discussed here: http://proceedings.mlr.press/v89/suggala19a/suggala19a.pdf and the cited paper [9]. This aspect does not negatively affect this paper, becaue the paper's focus is mostly on "welldefinedness of events", yet it would be good to know how this can fit into this paper’s formal studies."
>
> Yes, our formulations of adversarial risk correspond to "corrupted-instance" risk [9] as opposed to the "error-region" risk [9] or the adversarial risk studied in Suggala et al. Our analysis fixing the measurability issues in section 4 can be applied to other notions of adversarial risk as well. However, our other contributions in sections 5, 6 and 7 are applicable to the "corrupted-instance" family of adversarial risk that is commonly studied in literature on adversarial examples [25, 34].
>
> > "Line 290: “Our minimax theorems establish such an equality.” But in the cited paper [31] the authors say "Then we demonstrate, in Section 4, the non-existence of a Nash equilibrium in the deterministic setting of this game". How is their result compatible with yours? Please clarify."
>
> The non-existence of a Nash equilibrium in the deterministic setting in [31] is established for a "regularized" version of adversarial risk, where the adversarial budget constraint $d(x,x')\leq\epsilon$ is replaced with a regularization term. Their result is inapplicable to the standard setting with the common formulation of adversarial risk found in [25, 34] without any regularization term and without any relaxations on the adversary's budget constraint. Our Nash equilibrium result holds in the standard setting.
>
> > "The letter “T” is used for both the probability weight and the transportation notation."
>
> > "The term “decision region” is used in line 46 which is not clear. It is clarified later in line 109, but it is confusing the first time around."
>
> > "Line 362: “transport characterization adversarial risk” I think you need to add “of” after “characterization"
>
> Thank you again for identifying the ambiguity on the notation "T" for transport map and probability weight, and for catching the errors on Lines 46 and 362. We will fix these.

---

> > ### Comment · Reviewer_nVTq · 2021-08-17
> > **Thanks for clarifications**
> >
> > Thanks for the comments. They very helpful!
> >
> > Just some points:
> >
> > > Measurability may not be a problem, but unclear definitions certainly can be.
> >
> > Totally agreed, but it is also important to note that when the distributions are discrete, many previous definitions are trivially equivalent, and they only start to diverge when we go beyond discrete distributions. So, in a sense, in a world where we only care about discrete distributions, these issues do not arise. Having said that, as I said before, I see merit in doing a careful study in the non-discrete setting as well, and that is why I recommended acceptance.
> >
> > > Understanding binary classification is the first step to understanding adversarial robustness in more complicated cases.
> >
> > Again true. But it is also important to know to what extent the results of the paper automatically extend to non-binary settings. It is not fair to leave to them to the reader, and simply ask them to wait for a next paper! I am happy to see that in response to other reviews you said you will start by at least formulating the problem in the more general setting. I also agree that this would be a positive step.
> >
> > With regard to the misclassification issue (i.e., error region vs. corrupted instance definitions) I understand your point. I agree that there is interest in understanding "corrupted instance" as well, and in fact there are works studying it even before "adversarial examples" become a hot topic! But, I think it would be extremely helpful if you comment on what exact version you work with, as otherwise many might think that your work also implies "misclassification" of adversarial examples, while as you also agree, it does not in general.
> >
> > > The non-existence of a Nash equilibrium in the deterministic setting in [31] is established for a "regularized"...
> >
> > Thanks for clarifying this as well. I think it would be helpful to add such comments to the paper.
> >
> > Finally, I understand that you might not be able to fit all the full discussions in the camera ready of the paper (in whatever venue it appears), but you can always do so in the full version that will eventually be accessible as well (at least I hope such version would exist!). So I also encourage you to include many useful comments that you shared with us here, in the paper as well!

---

> > > ### Author Response · Authors · 2021-08-19
> > > **Thank you for your response.**
> > >
> > > Thank you for your kind and valuable comments. Below, we discuss them in brief.
> > >
> > > > "Totally agreed, but it is also important to note that when the distributions are discrete, many previous definitions are trivially equivalent, and they only start to diverge when we go beyond discrete distributions. So, in a sense, in a world where we only care about discrete distributions, these issues do not arise. Having said that, as I said before, I see merit in doing a careful study in the non-discrete setting as well, and that is why I recommended acceptance."
> > >
> > > Thank you for seeing merit in our analysis. It is indeed true that the definitions for $R_\epsilon$ and $R_{\oplus\epsilon}$ are trivially equivalent in the discrete setting. However, their equivalence with $R_{F_\epsilon}$ and $R_{\Gamma_\epsilon}$ is non-trivial even for discrete distributions. Because of this, our results in sections 5, 6 and 7 are non-trivial even in the discrete setting.
> > >
> > > > "Again true. But it is also important to know to what extent the results of the paper automatically extend to non-binary settings. It is not fair to leave to them to the reader, and simply ask them to wait for a next paper! I am happy to see that in response to other reviews you said you will start by at least formulating the problem in the more general setting. I also agree that this would be a positive step."
> > >
> > > We will update our discussion in section 8 with comments on which of our results extend naturally to non-binary settings and possible roadblocks for extending other results. We will also update section 3 with a presentation on adversarial risk for more general loss function and how it relates to our setting.
> > >
> > > > "With regard to the misclassification issue (i.e., error region vs. corrupted instance definitions) I understand your point. I agree that there is interest in understanding "corrupted instance" as well, and in fact there are works studying it even before "adversarial examples" become a hot topic! But, I think it would be extremely helpful if you comment on what exact version you work with, as otherwise many might think that your work also implies "misclassification" of adversarial examples, while as you also agree, it does not in general."
> > >
> > > W agree. We will update our discussion in section 3 to make it clear that the formulations we study in this paper belong in the "corrupted-instance" family of adversarial risks.
> > >
> > > We will include our comment on Nash equilibrium result in [31] to our discussion on related work in section 3.
> > >
> > > > "Finally, I understand that you might not be able to fit all the full discussions in the camera ready of the paper (in whatever venue it appears), but you can always do so in the full version that will eventually be accessible as well (at least I hope such version would exist!). So I also encourage you to include many useful comments that you shared with us here, in the paper as well!"
> > >
> > > We are glad that you found our comments useful. We will definitely add them to the full version of our paper.

---

### Official Review · Reviewer_9nkD · 2021-07-20

**Rating:** 7
**Confidence:** 3

**Summary:**

This paper formalizes different definitions of adversarial risk for binary classification and studies the equivalence between those definitions. Moreover, the authors analyze optimal adversarial risk through the lens of optimal transport, which leads to the existence of a Nash equilibrium of the adversarial robustness game. The contribution of this paper is theoretical.

**Limitations And Societal Impact:**

The authors discuss some of the limitations of their results in section 8. Comparing to the condition of Lebesgue measurable, I'm more curious about the case of general loss functions.

**Main Review:**


**Originality**

This paper is original and novel as far as I can see. It seems that one of the closest work to this paper is *M. S. Pydi and V. Jog. Adversarial risk via optimal transport and optimal couplings.* The authors discuss their advances from this one.

**Quality**

I did not notice wrong statements in the main paper. However, I only read the proofs in the main paper.

My questions are the following.

1.	This paper defines the adversarial risk as various set functions, i.e., loss function w.r.t. the decision region $A$. However, it is more general to define the adversarial risk w.r.t. the model $f$ itself.  For example, given the input space $\mathcal{X}$, output space $\mathcal{Y}$, the data generating distribution $p\in \mathcal{P}({\mathcal{X}\times \mathcal{Y}})$, and a cost function $c: \mathcal{Y} \times \mathcal{Y} \to [0,+\infty)$, the adversarial loss function can be modeled by $$R(f)=\int \sup_{x'\in B_\epsilon(x)} c(f(x'),y)\ \ \mathrm{d}p(x,y).$$
For binary classification, any function $f:\mathcal{X}\to \mathbb{R}$ defines a decision region by $A=f^{-1}((0, +\infty))$. By defining $c(\cdot,\cdot)$ to be the $\ell_{0/1}$ loss based on the decision region, it seems that it recovers the $R_{\oplus \epsilon}$ loss. Moreover, the above $R(f)$ can be applied to other settings, e.g., multi-class classification and regression.

    Denoting $\ell^\epsilon(x, y)=\sup_{x'\in B_\epsilon(x)} c(f(x'),y)$, to study the well-definedness for $R(f)$, it becomes to study the measurability of $\ell^\epsilon$ and the integrability of $\ell^\epsilon$ w.r.t. the data generating measure $p$. Therefore, does studying the well-definedness of $R(f)$ give the well-definedness of the set-based risk function? Moreover, what are the benefits of studying the set-based risks over the function based risk $R(f)$?

2. For Theorem 6.1, it is mentioned that the infimum on the right hand side is attained. I'm curious that is the supremum on the left hand side attained? It seems that it leads to whether the optimal $A$ in Theorem 6.2 is attained.
3. For non-theory people, the contribution of this paper may be limited. For example, an empirical person may think that the measurability is never a problem in real-world machine learning. Since NeurIPS is a big conference where attenders have diverse background, is there any point of this paper that may intrigue an empirical person?

**Clarity**

This paper is rigorous and well-written as far I see.

**Significance**

It is important in the sense that this paper rigorously defines several adversarial risk functions, filling the gap of previous work and building the ground for future work.

**Time Spent Reviewing:**

7

---

> ### Author Response · Authors · 2021-08-09
> **On extension to general loss functions, practical utility and other comments**
>
> Thank you for your thoughtful and encouraging review. We are glad that you found our paper to be "original and novel", "rigorous and well-written" and "filling the gap of previous work and building the ground for future work". Below, we answer your questions.
>
> ## On Extension to General Loss Functions
>
> > "This paper defines the adversarial risk as various set functions, i.e., loss function w.r.t. the decision region $A$. However, it is more general to define the adversarial risk w.r.t. the model $f$ itself..."
>
> Indeed, we already have results for general loss functions that we chose not to include in this submission. This was for two reasons: First was the limited manuscript length, and second was that not all results permit extensions to general loss functions (there is no analogue of Strassen's beyond 0-1 loss). So the 0-1 loss function presentation forms a complete story in itself that contains all the main ideas for extensions to general loss functions.
>
> > "Denoting $\ell^\epsilon(x, y)=\sup_{x'\in B_\epsilon(x)} c(f(x'),y)$, to study the well-definedness for $R(f)$, it becomes to study the measurability of $\ell^\epsilon$ and the integrability of $\ell^\epsilon$ w.r.t. the data generating measure $p$. Therefore, does studying the well-definedness of $R(f)$ give the well-definedness of the set-based risk function?"
>
> We emphasize that the 0-1 loss case may not be thought of as a special case of the general loss function. This is because the adversarial risk for general loss as defined in [25, 34] uses classifiers parametrized by some vector $w\in \mathbb{R}^p$. Our setting is *non-parametric* where the decision region of a classifier can be any set $A$ and challenging because notions such as continuity and convexity do not extend to set-functions easily. In fact, going from 0-1 loss to general loss is easier, which is not surprising considering the usual flow of "indicator functions->simple functions->measurable functions" used in analysis. As noted above, we do understand the well-definedness of risk for general loss functions as well, but chose to limit our discussion to the 0-1 loss for clarity and readability.
>
> > "Moreover, what are the benefits of studying the set-based risks over the function based risk $R(f)$?"
>
> Set-based risk gives us information-theoretic (i.e. algorithm independent) characterizations of adversarial risk. Furthermore, results from set-based risks permit extensions to function-based risk as well.
>
> ## On the attainment of Supremum over the set A
>
> > "For Theorem 6.1, it is mentioned that the infimum on the right hand side is attained. I'm curious that is the supremum on the left hand side attained? It seems that it leads to whether the optimal  in Theorem 6.2 is attained."
>
> We are not able to guarantee that the supremum over A is attained in Theorem 6.1 (analogously in Theorem 6.2). The reason for this is that the cost function $c_\epsilon$ that appears on the optimal transport problem in Theorem 6.1 is not continuous.
>
> ## On the Utility of our results to Empirical Research:
>
> > "For non-theory people, the contribution of this paper may be limited. For example, an empirical person may think that the measurability is never a problem in real-world machine learning. Since NeurIPS is a big conference where attenders have diverse background, is there any point of this paper that may intrigue an empirical person?"
>
> Measurability may not be a problem, but unclear definitions certainly can be. Indeed, our paper already shows the benefits of building a solid foundation for empirical researchers:
> 1. Algorithms for set-based risk can be very different from those for $W_\infty$-ball risk, but our equivalency results show that the desired robustness is achieved by both routes.
> 2. Our minimax results show adversarial classifiers are simply Bayes classifiers for the worst case distributions. Could one design algorithms that identify the worst-case distributions? This would be an entirely new approach to getting robust algorithms.
> 3. Our connection to the theory of capacities could allow for extensions to diverse classes of adversaries, such as total variation/Huber's/Prokhorov distance contaminations.

---

> > ### Comment · Reviewer_9nkD · 2021-08-17
> > **Thanks For The Response**
> >
> > Thanks for the response! I encourage the authors to include more discussions in the paper about the connections to general loss functions and empirical research. I keep my score unchanged, and I look forward to the results for general loss functions!

---

> > > ### Author Response · Authors · 2021-08-19
> > > **Thank you**
> > >
> > > Thank you for your comment. We will update section 3 with a discussion on adversarial risk for general loss functions and how it relates to our 0-1 loss setting. We will also update section 8 with a discussion on the utility of our results for empirical research.

---

### Author Response · Authors · 2021-08-09
**Thanks to All Reviewers + Answers to Common Concerns**

We thank all the reviewers for their insightful comments. We are glad that multiple reviewers found our work to be "novel", "rigorous" and "well-written". We are pleased that *all* the reviewers had something positive to say about our work, and that a *majority* of reviewers state that our work brings new "insights" on adversarial risk. Below, we address the comments raised by multiple reviewers.

- **"Why focus on measure theoretic issues?"**

We emphasize that rectifying measure theoretic issues with existing formulations of adversarial risk is one of our contributions, but NOT the main focus of our paper. We start our presentation with fixing measurability and well-definedness (in section 4) because otherwise we will not be able to rigorously present our main results in the subsequent sections, namely: relation to robust hypothesis testing and Choquet capacities in section 5, generalizing the results of [32,2] in section 6, proving minimax theorems and existence of Nash equilibria and extending the results of [27,31,3] in section 7.

- **"Measurability is not a problem in practice." / "How can applied research benefit from this work?"**

Measurability may not be a problem, but unclear definitions certainly can be. As we state in Lines 39-40, this has led to incorrect proofs and insufficient assumptions in some theoretical works. Hence, a mathematically rigorous foundation for adversarial risk is essential for future research. In addition to the importance of our work for theoretical research, our paper already shows the benefits of building a solid theoretical foundation for empirical researchers:
1. Algorithms for set-based risk can be very different from those for $W_\infty$-ball risk, but our equivalency results show that the desired robustness is achieved by both routes.
2. Our minimax results show adversarial classifiers are simply Bayes classifiers for the worst case distributions. Could one design algorithms that identify the worst-case distributions? This would be an entirely new approach to getting robust algorithms.
3. Our connection to the theory of capacities could allow for extensions to diverse classes of adversaries, such as total variation/Huber's/Prokhorov distance contaminations.

- **Readability for a broader audience**

We will take the following steps to make our paper more readable to a broader audience.
1) Since the adversarial risk definition with general loss function is more familiar to applied researchers, we will introduce the discussion in section 3 with such a definition and then specialize to 0-1 loss.
2) We will add figures to better illustrate various definitions presented in section 3.

- **"Corrupted-instance" risk vs "error-region" risk**

All the "many faces" of adversarial risk that we present in this paper correspond to "corrupted-instance" family of adversarial risks. Our goal in this paper was not to provide an exhaustive survey of all the notions of adversarial risk, but rather to study various formulations of a specific notion of adversarial risk and its connections to optimal transport, game theory and Choquet capacities. We will state this explicitly in section 3, and mention the other notions of adversarial risk not discussed in our paper.

In addition to the above comments, we have provided detailed responses to all the comments of the reviewers individually. Again, we appreciate all your time and effort put into reviewing our work.

---

### Decision · Program_Chairs · 2021-09-27

**Decision:**

Accept (Poster)

**Comment:**

All the reviewers agreed that this paper provides important contributions in the area of robust learning. I recommend to the authors to incorporate all the comments made by the reviewers in the updated version of the paper. The reviews have indeed brought many important points.